# Advances, Challenges, and Future Perspectives of Microwave Reflectometry for Plasma Position and Shape Control on Future Nuclear Fusion Devices

**DOI:** 10.3390/s23083926

**Published:** 2023-04-12

**Authors:** Bruno Gonçalves, Paulo Varela, António Silva, Filipe Silva, Jorge Santos, Emanuel Ricardo, Alberto Vale, Raúl Luís, Yohanes Nietiadi, Artur Malaquias, Jorge Belo, José Dias, Jorge Ferreira, Thomas Franke, Wolfgang Biel, Stéphane Heuraux, Tiago Ribeiro, Gianluca De Masi, Onofrio Tudisco, Roberto Cavazzana, Giuseppe Marchiori, Ocleto D’Arcangelo

**Affiliations:** 1Instituto de Plasmas e Fusão Nuclear, Instituto Superior Técnico, Universidade de Lisboa, 1049-001 Lisboa, Portugal; 2Max-Planck-Institut für Plasmaphysik, Boltzmannstr. 2, D-85748 Garching, Germany; 3Institut für Energie- und Klimaforschung, Forschungszentrum Jülich GmbH, D-52425 Jülich, Germany; 4Institut Jean Lamour, UMR 7198 CNRS-Université de Lorraine, BP 50840, F-54011 Nancy, France; 5Consorzio RFX, 35127 Padova, Italy; 6ENEA, Fusion and Technologies for Nuclear Safety Department, C.R. Frascati, Via E. Fermi 45, 00044 Frascati, Italy

**Keywords:** microwave antennas, microwave propagation, millimetre wave propagation, microwave circuitry, millimetre wave circuitry, microwave measurements, plasma diagnostic, fusion plasma

## Abstract

Providing energy from fusion and finding ways to scale up the fusion process to commercial proportions in an efficient, economical, and environmentally benign way is one of the grand challenges for engineering. Controlling the burning plasma in real-time is one of the critical issues that need to be addressed. Plasma Position Reflectometry (PPR) is expected to have an important role in next-generation fusion machines, such as DEMO, as a diagnostic to monitor the position and shape of the plasma continuously, complementing magnetic diagnostics. The reflectometry diagnostic uses radar science methods in the microwave and millimetre wave frequency ranges and is envisaged to measure the radial edge density profile at several poloidal angles providing data for the feedback control of the plasma position and shape. While significant steps have already been given to accomplish that goal, with proof of concept tested first in ASDEX-Upgrade and afterward in COMPASS, important, ground-breaking work is still ongoing. The Divertor Test Tokamak (DTT) facility presents itself as the appropriate future fusion device to implement, develop, and test a PPR system, thus contributing to building a knowledge database in plasma position reflectometry required for its application in DEMO. At DEMO, the PPR diagnostic’s in-vessel antennas and waveguides, as well as the magnetic diagnostics, may be exposed to neutron irradiation fluences 5 to 50 times greater than those experienced by ITER. In the event of failure of either the magnetic or microwave diagnostics, the equilibrium control of the DEMO plasma may be jeopardized. It is, therefore, imperative to ensure that these systems are designed in such a way that they can be replaced if necessary. To perform reflectometry measurements at the 16 envisaged poloidal locations in DEMO, plasma-facing antennas and waveguides are needed to route the microwaves between the plasma through the DEMO upper ports (UPs) to the diagnostic hall. The main integration approach for this diagnostic is to incorporate these groups of antennas and waveguides into a diagnostics slim cassette (DSC), which is a dedicated complete poloidal segment specifically designed to be integrated with the water-cooled lithium lead (WCLL) breeding blanket system. This contribution presents the multiple engineering and physics challenges addressed while designing reflectometry diagnostics using radio science techniques. Namely, short-range dedicated radars for plasma position and shape control in future fusion experiments, the advances enabled by the designs for ITER and DEMO, and the future perspectives. One key development is in electronics, aiming at an advanced compact coherent fast frequency sweeping RF back-end [23–100 GHz in few μs] that is being developed at IPFN-IST using commercial Monolithic Microwave Integrated Circuits (MMIC). The compactness of this back-end design is crucial for the successful integration of many measurement channels in the reduced space available in future fusion machines. Prototype tests of these devices are foreseen to be performed in current nuclear fusion machines.

## 1. Introduction

The International Thermonuclear Experimental Reactor (ITER) in France and the conceptual design of a demonstration power plant (DEMO) are advancing the development of nuclear fusion as a viable solution for large-scale energy production. In these reactors, nuclear fusion reactions occur when a deuterium and tritium (D-T) plasma is heated to temperatures ten times higher than at the core of the Sun. The position control of this extremely hot plasma inside the reactor’s fusion chamber is one of the most critical issues in the operation of these power-generating devices. As such, real-time feedback control of the plasma position plays a vital role in machine protection and disruption avoidance, being crucial for successful reactor operation. However, controlling plasma parameters in future reactor-grade fusion tokamaks, such as ITER and DEMO, presents significant challenges. During the ramp-up phase, it is crucial to prevent the plasma from impinging the inner vessel walls and to avoid destructive disruptions during steady-state operation. Presently, magnetic measurements are used for plasma control, but in future long pulse tokamak devices, such as ITER and DEMO, these measurements may be impacted by drifting integrators, radiation-induced voltages, or radiation damage to the magnetic pickup coils. These effects could compromise the magnetic equilibrium reconstruction, leading to premature discharge termination or damage to plasma-facing components.

Several modern measurement diagnostics used in fusion rely on Radiofrequency (RF) techniques to probe the plasma. The low amplitude of the probing waves introduces negligible perturbations to the plasma. However, the interaction between the electromagnetic (EM) field of the propagating waves and the magnetized plasma can cause changes in the amplitude, phase, polarization state, and spectrum of the waves due to propagation close to the plasma layers where cut-off reflection occurs. The location of the reflecting layers can be deduced by measuring the delay caused by the round-trip of the reflected waves. This information is used to reconstruct the electron density profile, which enables the real-time location of the separatrix—the last closed magnetic surface—provided a good estimation for its density is known.

Microwave reflectometry is a radar-based technique that can be used to determine the radial distribution of plasma density in fusion experiments. This technique has been proposed as an alternative to magnetic measurements for plasma position control. The first proof of principle for reflectometry-based plasma position feedback control was successfully demonstrated on ASDEX Upgrade [1]. Microwave reflectometry is a well-proven technique in the plasma fusion community and is used for a variety of physics measurements. One application is the study of turbulence characteristics in fixed electron density layers by continuously probing the plasma with a fixed frequency wave. Other uses for microwave reflectometry can also be found in the literature [2].

## 2. Probing Plasmas with Microwave Reflectometry

The Maxwell equations and the motion of plasma particles induced by the EM waves can be used to describe the propagation of EM waves in an inhomogeneous medium, such as a fusion plasma. This description is possible if we assume [3] that: First, when there are no EM perturbations, the plasma particles remain in their equilibrium positions (known as the cold plasma approximation). Second, the plasma is treated as a two-fluid medium consisting of ions and electrons that are coupled through the EM field of the wave (known as the fluid approximation). Third, it is considered that only the electrons contribute to the medium polarisation because the frequency of the EM waves is much higher than the eigen ion plasma frequencies, cyclotron $f_{ci}$ and ion plasma $f_{pi}$ frequencies (high-frequency approximation). Lastly, the plasma is considered homogeneous in all directions except for the direction of wave propagation (known as the slab approximation). This approach allows for a comprehensive understanding of the behaviour of EM waves in fusion plasma.

Suppose the former approximations are assumed for a wave propagation direction perpendicular to the plasma magnetic field (k⊥B), where k is the propagation direction and B is the plasma magnetic field, two propagation modes can be identified [4]: the ordinary mode (O-mode), where the wave’s electric field is parallel to the plasma magnetic field (E∥B), as illustrated in Figure 1, and the extraordinary mode (X-mode), where the electric field of the wave is perpendicular to the magnetic field of the plasma (E⊥B). For the case of a tokamak plasma, the poloidal component of the magnetic field is negligible when compared to the toroidal component ($B_{\varphi }\gg B_{\theta }$), and therefore, only the toroidal component of the magnetic field is considered, i.e., $B≃B_{\varphi }$.

The refractive index, $N_{O}$, for an O-mode wave with frequency f is described by [5]:(1)$$N_{O}={\left[1-{\left(f_{pe}/f\right)}^{2}\right]}^{1/2}={\left[1-n_{e}/n_{co}\right]}^{1/2}$$
where $n_{e}$ is the electron density and $f_{pe}={(1/2\pi )\left(n_{e}e^{2}/\epsilon_{0}m_{e}\right)}^{1/2}$ is the electron plasma frequency. Wave propagation can occur for $f>f_{pe}$ or for $n_{e}<n_{co}$, where $n_{co}$ is the cut-off density given by:(2)$$n_{co}={\left(2\pi f/e\right)}^{2}\epsilon_{0}m_{e}$$
where e is the electron charge, $m_{e}$ is the electron mass and $\epsilon_{0}$ is the permittivity of free space.

A microwave reflectometer is a short-range radar dedicated to the plasma measurement: a wave with frequency f is launched at normal incidence into the plasma, propagating through it until it reaches the cut-off layer with density $n_{co}(f)$, for O-mode polarization, where the refractive index becomes zero and the wave is reflected. The position of the density layers, at which the corresponding reflections occur, can be obtained by sweeping the frequency of the incident probing waves and measuring the waves’ group delay due to the round trip from the launching antenna to the plasma and back to the receiving antenna. This information, plus an initial cut-off position for the lowest probing frequency, can then be used to calculate the plasma density profile along the reflectometer’s line-of-sight (LOS). Frequency-modulated continuous wave (FMCW) reflectometry using ordinary (O-mode) plasma waves can measure the radial electron density profile independently of any parameters other than the plasma density ($n_{e}$) since O-mode propagation depends solely on the electron density distribution. For that reason, reflectometry measurements have been proposed on ITER to complement the standard magnetic measurements that are used in present machines for plasma position control [6] during the steady-state operation periods. In addition, the reflectometry diagnostic displays significant advantages, namely its minimum access requirements and compatibility with the harsh environment of a fusion device, plus the capability to perform density profile measurements in a short time scale (on the order of a few µs) inherent to a radar technique. Although for control purposes, measurements of the edge density profile are not an identical replacement for magnetic measurements, under certain conditions, they can provide similar capabilities [7]. In current fusion devices, position control is accomplished by changing the currents in the poloidal field coils based on the real-time estimation of the plasma shape and, in particular, the estimation of the plasma magnetic separatrix. Because the separatrix results exclusively from the reconstruction of the magnetic flux distribution, there is no direct relation with the electron density. Thus, a good estimation for the density just inside this flux surface is needed to track the gap between the separatrix and the vessel’s first wall using a radial density profile given by reflectometry. A fusion plasma is a turbulent medium, and obtaining reliable measurements from waves propagating in it has proven to be a difficult and complex task. Before considering this technique as a reliable alternative for measuring plasma position, multiple challenges must be addressed. This article outlines the necessary advancements required to integrate this radar-like technique into the control systems of a fusion machine.

## 3. Demonstration of the PPR Principle at ASDEX-Upgrade

The use of microwave reflectometry for position control was successfully demonstrated for the first time on the ASDEX Upgrade (AUG) tokamak [1]. To control the plasma column position, the location of the outer plasma boundary was tracked in real-time (RT)—1 ms measurement rate/1 ms control loop cycle duration—using a custom-built very high bandwidth streaming data acquisition system [8], an optimized digital signal processing analysis and a dedicated neural-network based profile reconstruction algorithm [9]. The approach combined the RT reflectometry edge profile and an RT scaled line integrated density measurement from interferometry. After the first successful demonstration of plasma position control using reflectometry, the diagnostic’s hardware was updated to acquire a higher number of signals and to improve its RT data-acquisition and data-processing capabilities. The improvements were aimed at producing a second demonstration of a plasma position control using both AUG’s equatorial reflectometers, probing the tokamak high (HFS) and low field sides (LFS) simultaneously. This upgrade also aimed to achieve a fourfold increase of the system RT measurement rate (250 μs measurement rate/1 ms control loop cycle duration) and to integrate the RT reflectometry diagnostic in the AUG’s Discharge Control System (DCS), a solution closer to ITER’s PPR foreseen operation mode [10]. Both successful position control demonstrations were performed in ITER-relevant plasma regimes and configurations, providing a decisive contribution to the design of future PPR control systems.

The AUG broadband reflectometry diagnostic started operation at the beginning of the 1990’s. The system was designed by IPFN/IST researchers and is also operated by researchers from the research unit in close collaboration with the AUG Team of the Max-Planck Institut für Plasmaphysik (IPP). For the plasma position control experiments, two FMCW O-mode reflectometry systems [7] were used to probe the plasma, installed at the tokamak equatorial plane, from the high (inner) and low (outer) magnetic field sides (HFS and LFS, respectively) of the device (see Figure 2). The HFS reflectometers operate in the 18–75 GHz frequency range, using bands K (18–26.5 GHz), Ka (26.5–40 GHz), Q (33–50 GHz), and V (50–75 GHz). In the LFS, this range is extended, up to ∼110 GHz, with an additional reflectometer operating in the W band (75–110 GHz). In the LFS, two additional channels operating in X-mode are used to cover the 35–75 GHz range (using bands Q and V). The FMCW microwave reflectometry system on ASDEX Upgrade uses fundamental rectangular waveguides connected to hog-horns with elliptical mirrors. Their orientation sets the electrical field of the probing beam in the toroidal direction. The ASDEX Upgrade magnetic pitch angle is in the order of 10° [11], which implies that 97% of the probing signal is sent in O-mode. The 3% injected in X-mode are attenuated in the waveguide. Therefore, no calibration procedure is required to set the correct polarization. If circular waveguides were used, the correct polarization would require calibration using a mirror with a grid. The original system design was heavily influenced by the limited space available within the tokamak to route the waveguides and position the antennas behind the vessel heat shield. This shield, consisting of heat-resistant tiles, protects the inner vessel walls and plasma-facing components. The constraints, along with the experience with reflectometers built to the old ASDEX tokamak, resulted in an optimized design using mono-static focused hog-horn antennas (emission and reception are made using a single antenna), where the reception is always optimized with moderated directivity. This configuration allows for a more compact solution, especially in the HFS region, where space is more restricted. The system proved to cope quite well with rather large vertical displacements in the order of 20 cm [12]. The higher frequency bands, V and W, which probe plasma layers further away from the antennas, were later improved to increase the system’s Signal to Noise Ratio (SNR) by implementing a heterodyne detection scheme with a phase-locked-loop (PLL) and a second oscillator to drive a harmonic mixer to down-convert the reflected signals to an intermediate frequency (IF). In these channels, the IF signal is still detected with a Schottky-diode detector at approximately 1 GHz. O-mode full-wave 1D simulations were performed to assess measurement sensitivity to plasma turbulence and initialization of the non-probed plasma density range ($0\leq n_{e}\leq 0.36\times 10^{19}$ $m^{-3}$) [13]. These simulations demonstrated that the spatial resolution always remains below 5 mm. Due to its design, the AUG reflectometry system remains the only broadband diagnostic worldwide capable of directly probing the plasma from the machine’s HFS.

The usual method for controlling the position of the plasma volume is to monitor the location of a few specific control points of the magnetic separatrix (green line on the plot of Figure 2). Then, the position controller changes the configuration of the magnetic fields that confines the plasma on the estimation of their locations ($R_{in},R_{out},Z_{I}$). For reflectometry to serve as a replacement for magnetic diagnostics, it must provide the controller with the same input information accurately, within an 1 cm error bar. In the position feedback control demonstrations performed at AUG, reflectometry provided RT estimates for $R_{in}$ and $R_{out}$, while $Z_{I}$ remained magnetics based. The developed systems and real-time algorithms’ performance was evaluated by conducting dedicated discharges, where the magnetic and reflectometry-based controllers were switched. The controller demonstrated its ability to handle transitions between the two input sources (magnetic and reflectometric) and maintained the controlled position close to the programmed trajectories throughout the process. During the demonstration of plasma position control using both HFS and LFS reflectometers, the estimates of the inner, $R_{in}$, and outer, $R_{out}$, separatrix positions were combined to estimate the geometric plasma radius, $R_{geo}=(R_{in}+R_{out})/2$, which replaced the corresponding input magnetic measurement in the position controller.

The experiment was performed during the high-confinement mode (‘H-mode’) in the presence of Edge Localized Modes (ELMs) instabilities which results in bursts of energy and particles at the plasma edge. Electron density profiles measured by reflectometry can be severely affected by the ELMs, and it was required to develop algorithms to automatically validate reflectometry measurements to cope with the effects of these cyclic transitory periods. Figure 3 shows the main time evolution of the measured parameters of one of the four discharges used in the demonstration. The top plot shows the line integrated density (H-1) at the equatorial plane, Deuterium (D) fuelling, neutral beam power (NBI), ECRH heating power, and plasma current ($I_{pa}$). During the ELMy H-mode flat-top phase, the position control switched to the reflectometry-based controller from t∼2.6 s until t∼7.6 s. After that moment, the ramp down of the plasma is performed by the magnetic controller. For the period where reflectometry was used to control the plasma, the geometric radius of the plasma column was programmed (CTRL trace on the bottom plot of Figure 3) to swing 1.5 cm (non-symmetrically) around its original position. The reflectometry-based controller maintained the reflectometry’s $R_{geo}$ within ∼±0.5 cm of the target trajectory. The reflectometry estimates for $R_{in}$, $R_{out}$, and $R_{geo}$ are coherent with their magnetic counterparts, demonstrating very good precision although with improvable accuracy: ∼1.5 cm and ∼2 cm offsets to the magnetics at the LFS and HFS, respectively. The corresponding $R_{geo}$ input offset was successfully handled by the position controller when switching to and from reflectometry input at t∼2.6 s and t∼7.6 s. Similarly, to the original LFS-only demonstration, in these experiments, the system operated flawlessly during the programmed reflectometry control phases, and its performance was reproducible in all four discharges.

The success of these experiments proved that it would be possible to fulfil the ITER requirements for plasma control using microwave reflectometry as an alternative to standard magnetic measurements. The use of a reflectometry-based plasma position control can contribute to achieving the high availability and reliability levels required for the operation of ITER, helping to pave the way toward the realization of future fusion power plants.

## 4. On the Integration of a PPR System at COMPASS

The COMPASS tokamak, located in IPP-CAS Prague, is a compact divertor device that features an ITER-relevant geometry and can operate in H-mode, which makes it an ideal platform for conducting pedestal and edge physics studies relevant to ITER. COMPASS has installed a microwave reflectometer, operating in O-mode, and has a real-time control system based on the Multi-Threaded Application Real-Time executor (MARTe). These two features create optimal conditions for the further development and advanced demonstration of a reflectometry-based PPR system. To implement the PPR system at COMPASS [14], it was necessary to seamlessly integrate the microwave reflectometry diagnostic with the Real-Time Control system (RTCS) [15,16], which is responsible for the control of the plasma position based on standard magnetic signals. Although the COMPASS reflectometry channels can be swept in <10 μs, the limited upload streaming capabilities of the system’s ATCA acquisition system and the consequent reflectometry RT measurement latency conditioned the implemented PPR to be used in COMPASS’s slow 500 μs control loop cycle.

COMPASS O-mode reflectometry diagnostic [17] probes the plasma through a quartz vacuum window located at a midplane LFS port. K, Ka, and U microwave bands are used to probe plasma electron densities in the $0.8-4.3\times 10^{19}$ $m^{-3}$ range. Two separate crates are used to enclose the microwave hardware and the control electronics (one for the K and Ka bands and the other for the U band). The system is also ready for an E band upgrade in the future. Each crate has a controller board consisting of a PC/104 embedded computer with additional electronics to interface and drive the microwave oscillators. Its digital electronics are synchronized with a local high-precision 10 MHz time-base synchronization clock. The diagnostic can operate at fixed, hopping, or swept frequency modes. The controllers allow for the remote operation of the diagnostic via the TCP/IP protocol over Ethernet, with a set of proprietary commands. The microwave electronics of each band provide the single-sideband (SSB) modulation, the in-phase (I), and the quadrature (Q) detection where the SSB signal has the highest bandwidth of the set, corresponding to an IF of 48 MHz for the K and Ka bands and 96 MHz for the U band. The profile measurements used by the PPR system are obtained by sweeping all bands simultaneously using a configurable sweep duration; an external trigger source sets the sweep repetition rate.

Figure 4a shows the real-time reflectometry profiles for PPR control demonstration discharge 19691 (circular L-mode discharge). Figure 4b shows the separatrix location estimated from the reflectometry profiles (green) from which the PPR magnetic radius controller input was derived. During the period highlighted in grey, ∼140 ms, the estimated magnetic radius was used for the radial position controller input (red), replacing the actual magnetic measurement (blue) in the slow controller (500 μs control loop cycle time). In the case of COMPASS, the separatrix location had to be inferred from the profiles using a pre-validated fixed density, representative of the plasma regime, unlike in AUG, where a fraction of the average density measured in real-time was used.

The O-mode microwave reflectometer was successfully integrated into the existing real-time diagnostic network system. This integration enabled the use of the diagnostic for position feedback experiments by providing the MARTe main controller with a PPR-based radial position estimation. The well-defined software block structure of the MARTe framework largely expedited the design, development, and commissioning of the real-time application. The implementation of PPR at COMPASS made it possible to extend its applicability limits by demonstrating that it can cope with twice as fast control dynamics as in the original demonstrations performed in AUG.

## 5. Plasma Position Reflectometry at ITER

The ITER Plasma Position Reflectometry (PPR) system was planned to play a supplementary role to magnetic diagnostics in providing information to the ITER Plasma Control System (PCS) about the gap between the plasma and the first wall. Although the project was descoped in 2020, a considerable amount of work has been accomplished that paved the way to solve many of the similar (and worse) problems faced by the implementation of plasma position reflectometry in future fusion devices such as the Divertor Test Tokamak (DTT) and DEMO.

On ITER, the PPR system consisted of four FMCW O-mode reflectometers in full bi-static configuration covering densities up to ∼$7\times 10^{19}$ m^−3^ (15–75 GHz). These reflectometers were to be installed at four different locations, known as gaps 3, 4, 5, and 6, both in-vessel (gaps 4 and 6) and inside port plugs (gaps 3 and 5). The main goal of the diagnostic was to measure the edge density profile in real time with high spatial (<1 cm) and temporal (100 μs) resolutions. To reduce transmission losses, the system used oversized 20×12 mm rectangular waveguides inside the vacuum vessel and port plugs.

The antennas and part of the transmission lines of gaps 4 and 6 were to be installed inside the vacuum vessel, as depicted in Figure 5. Due to direct exposure to the plasma, these antennas and the 90° bends connecting to the transmission lines were expected to suffer significant loads that could compromise their integrity.

The system of gap 4 had the antennas located in the low-field side (LFS) of sector 9, between the blanket modules (BMs) in rows #11 and #12. The support of the 90° bend was bolted to ITER standard in-vessel attachments welded to the wall [18]. From the 90° bends, the microwaves were routed the feed-outs at the bottom of upper port 1 using straight and curved sections of the waveguide.

In the case of gap 6, the antennas were to be installed at the high-field side (HFS) of sector 7, between the BMs in rows #3 and #4. The support of the 90° bend would be attached to the vessel in the same way as in gap 4. The transmission line of gap 6 was then routed along the inner wall of the vacuum vessel between the bend and one of the feed-outs on the top of upper port 14. Further details about the design of the PPR system are provided in [19,20].

### 5.1. Thermal Loads and Material Testing

As mentioned, the in-vessel components of gaps 4 and 6, namely the antennas and 90° bends, would be subjected to significant thermal loads that could compromise their integrity. Figure 6 shows the expected temperature distributions in these two components for gap 6, which were estimated using ANSYS Mechanical by applying the load specifications for the PPR in-vessel components as per the ITER guidelines for thermal analyses. As shown, the maximum operating temperatures would be well above the limit of 450 °C for the selected material (ITER-grade stainless steel) under neutron irradiation. Thus, it was proposed that these components were manufactured from other materials, such as tungsten or a nickel-based superalloy [21].

To minimise the EM forces induced during plasma disruptions, the in-vessel waveguides were made from ITER grade stainless steel (SS 316(LN)-IG), but coated inside with a thin (15–25 μm) Copper layer to keep ohmic losses at a minimum. Prototypes of these waveguides were tested at the IPFN-IST microwave laboratory. The results obtained (see Figure 7) have shown that the stainless-steel, Cu-coated waveguides had similar performance to the theoretical predictions and to Cu-only waveguides, both in O-mode and X-mode-like polarizations. These results were reproducible for all the waveguide prototype samples tested.

### 5.2. Performance Assessment of Waveguides Bends

As expected, the performance impact of the in-vessel oversized waveguide bends of gaps 4 and 6 was a critical aspect of the PPR design, namely the 90° bends just behind the antennas and the 125° bend of gap 4. It is well known that these bends are prone to excite higher-order modes and create resonances, which can adversely affect performance if not designed carefully. The 90° and 125° bends of the in-vessel PPR systems were first studied via 3D EM simulations using both the frequency-domain solver of ANSYS HFSS and the time-domain solver of CST Microwave Studio (MWS).

Figure 8a shows the results obtained with HFSS for the optimization of the 125° bend, which was split into two identical 62.5° bends with hyperbolic secant geometry. The optimized shape (C9) clearly improved performance with respect to the constant radius bend (Baseline) across the 15–75 GHz frequency range (with just a small degradation < 0.2 dB visible above 70 GHz) while complying with the in-vessel space restrictions [22]. Figure 8b depicts the results obtained with HFSS for the optimized bend against the ones obtained using MWS, which exhibit a perfect match between both tools. The electric field distributions inside the optimized 125° bend (C9), as obtained with HFSS for 15 GHz, 45 GHz, and 75 GHz, are shown in Figure 9.

The 125° bend was prototyped to assess the EM performance of the optimized design and the manufacturing process, namely the copper-coating and the bending to the specified geometry. During the site acceptance inspection of the prototype samples, some problems were readily identified:All samples showed a lack of (copper) coating near the edge of the flanges, as shown in Figure 10a, as well as in some spots near the flanges, as observed in Figure 10b. The lack of copper was also visible inside the waveguides along the corners formed by the inner walls and in the walls themselves.Marks were visible in the inner surface of the bend, probably caused by some tool inserted in the bend during bending, as depicted in Figure 10c,d.local shape deformation was clearly visible in all samples around the middle of the bend, as shown in Figure 10e, where, per specification, there should be a straight section due to the connection between the two 62.5° hyperbolic-secant bends—as is visible in Figure 10f, the deformation extends to the inner wall of the waveguide, which anticipates performance issues.Figure 10e clearly shows that the received samples were not symmetric components as specified, and judging by the relative position of its flanges, it seems the samples were all slightly twisted around the propagation axis.Finally, measurements taken as part of the site acceptance tests have shown that the internal section of the received samples was about 0.2 mm (both in height and width) smaller than the specified dimensions.

The main goal of the tests was to measure the performance of the samples in the frequency range 15–75 GHz for the TE_01_ waveguide mode (O-mode-like polarization), as used by the PPR system. The testing frequency range was covered using the standard frequency bands K (extended down to 15 GHz), Ka, Q, and V. Waveguide tapers were used to enable the connection between the non-standard dimensions of the bend and the standard dimensions of the hardware equipment used in the tests. For each band, the measurements were preceded by a calibration for that particular band of the Vector Network Analyzer (VNA) used in the tests. Due to the small attenuations of the bend in certain frequency ranges, the difference to the reference measurements was within the resolution limit of the VNA (±0.1 dB). Thus, the measurements presented here were performed by activating the Secondary Match Correction (SMC) feature of the VNA, which improves the resolution of the measurements by slowing down the frequency sweep.

For each tested sample, the measured attenuation was compared with what can be expected from the analytical conductivity losses for the TE_01_ mode propagating through a straight rectangular copper waveguide, $\alpha_{c}$, given by
(3)$$\alpha_{c}\left[\mathrm{dB}/m\right]=8.686\frac{2R_{m}}{bZ_{0}\sqrt{1-k_{c}^{2}/k_{0}^{2}}}\left[\left(1+\frac{b}{a}\right)\frac{k_{c}^{2}}{k_{0}^{2}}+\frac{b}{a}\left(\frac{1}{2}-\frac{k_{c}^{2}}{k_{0}^{2}}\right)\right],$$
where, a and b, with a>b, are the waveguide cross-section dimensions, $k_{c}$ is the cut-off wave number, $k_{0}$ is the wave number corresponding to frequency ω=2πf, $Z_{0}$ represents the free-space impedance, and $R_{m}={\left(\omega \mu_{0}/2\sigma_{Cu}\right)}^{1/2}$, where $\mu_{0}$ is the free-space permittivity and $\sigma_{Cu}$ is the electric conductivity of copper.

The results obtained from the tests of the prototype samples (numbered 18110703 to 18110706 and 18110708) are depicted in Figure 11, along with the analytical attenuation of a straight 0.33 m-long rectangular waveguide with the same cross-section, as obtained from Equation (3). As can be observed for all bends, there are two frequency regions where the experimental attenuation deviates significantly from the analytical losses: between 43 GHz and 54 GHz, where the bends exhibit moderate attenuation (up to 2 dB), and above 62 GHz, where the attenuation rapidly increases from ∼0.5 dB up to ~10 dB at 75 GHz. Out of these regions, all samples exhibited good performance, with the attenuation remaining below 0.5 dB.

These results have shown that the tested bends were inadequate for the PPR system and were not a surprise given the observations made during the acceptance of the prototypes, which indicated that the (bending) process used to manufacture this component, although reproducible as demonstrated by the results of the tests, is clearly unable to provide the precision and accuracy required by the specified hyperbolic secant geometry, whose main goal is to mitigate the excitation of higher-order modes that are known to deteriorate performance. In fact, although not exactly coincident with the ones of the tested samples, the attenuations estimated with HFSS and MWS also exhibit two frequency regions where the performance of the bend deteriorates slightly with respect to the theoretical losses, which are shown to be mainly due to the excitation of the TE_02_ mode. This similarity indicated that the bad performance of the prototype could be due to mode conversion losses into higher order modes, which were not mitigated as expected due to the local deformations and overall poor shape accuracy of the manufactured prototype. One of the lessons learned for future developments was that the prototyping required to validate the EM design should be separated from the prototyping required to validate manufacturing processes to allow for a better understanding of the root causes of performance issues.

### 5.3. Avoiding Crosstalk in the In-Vessel Waveguide Flanges

The ITER in-vessel waveguides consisted of multiple sections of parallel waveguides connected using flanges. Therefore, crosstalk at the flanges was a concern. The envisaged solution to prevent crosstalk consisted of matching male-female flanges with a centre pole, as depicted in Figure 12a. A prototype flange was tested using the setup illustrated in (Figure 12b). First, a reference isolation measurement was performed with both ports of the VNA directly terminated by a matched load. Secondly, the crosstalk measurement (A→D) was performed with the flanges inserted, and B and C terminated using matched loads. The results of those tests are illustrated in Figure 13. As shown, the envisaged solution effectively reduced the flange crosstalk to negligible levels across the entire 15–75 GHz frequency range.

### 5.4. Testing the Integration of the Antennas of Gaps 4 and 6 between Blanket Modules

To optimize the design of the antennas for gaps 4 and 6, extensive simulations were performed using the REFMUL Finite-Difference Time-Domain (FDTD) code [23,24,25], which is able to consider the geometric characteristics of the space between BMs where the antennas would be installed. From these simulations, some concerns were raised regarding the effect of the blankets on the signals due to reflections and diffraction. Thus, prototypes of the antenna assembly, consisting of the antennas and feeding waveguides, were built and tested, both isolated and integrated into a mock-up of the blanket modules to simulate the effect of these structures. The prototypes of the antennas and the mock-up of the BMs for gap 6 are shown in Figure 14.

The radiation patterns of the antenna assembly were measured in an 3.8×2.5×2 m anechoic chamber with and without the BMs. Figure 15 shows the antenna inside the anechoic chamber during these measurements. Two different antennas were prototyped and tested: antenna #1 (baseline), consisting of two parallel pyramidal horns with a length of 115 mm, a toroidal flare of ±2 mm, and poloidal flare of ±1 mm, and antenna #2, consisting of two parallel pyramidal horns with a length of 115 mm, a toroidal flare of ±2 mm, and poloidal flare of ±4 mm.

To reduce the time required to complete the measurements, instead of full 360° scans, the radiation patterns were only measured in the range −120° to +120° (240° scans). For each frequency band, the radiation pattern of a standard calibrated horn antenna was measured to absolutely calibrate the radiation patterns of the Antenna Under Test (AUT). Before each scan, the phase centre of the AUT was aligned with the Probe Antenna (PA). In the measurements, both the AUT and the PA could be rotated in their vertical plane; the AUT was also able to be rotated in the horizontal plane. The measurement plane (E or H) was selected by rotating the AUT and PA antennas in the vertical plane as desired. For cross-polarization measurements, the AUT and PA antennas were rotated such that they were in complementary planes.

For the different bands, the radiation patterns were measured at the following discrete frequencies:Ku band: 15 GHz, 16.5 GHz.Ka band: 18 GHz, 19.7 GHz, 22.25 GHz, and 24.8 GHz.Ka band: 29.2 GHz, 33.25 GHz, and 37.3 GHz.Q band: 42 GHz, 45 GHz, and 48 GHz.V band: 55 GHz, 62.5 GHz, 70 GHz, and 75 GHz.

Depending on the frequency range, different setups were used in the anechoic chamber due to the need to keep the dynamic range of the measurements above 25 dB for all bands (the loss of dynamic range occurs because the output power from the Agilent Technologies N8361A VNA used in the measurements decreases as the frequency increases with the simultaneous increase of the losses on the high-frequency cables).

We expected the measurements to show the effects of the blanket modules to be most noticeable in the poloidal plane, which for the TE_01_ mode corresponds to the antenna’s H-plane, due to the fact that, with respect to the dimensions of the antennas, the aperture between blankets is sufficiently large in the toroidal plane (antenna’s E-plane) so that it does not significantly affect the radiation patterns in this plane. The E-plane co-polar radiation patterns measured for antennas #1 and #2 with and without the blanket modules in place confirmed those expectations showing that the modifications in the radiation patterns of both antennas were, in fact, not significant. For both antennas, the H-plane co-polar radiation patterns measured with and without the BMs are illustrated in Figure 16.

As can be observed, the most noticeable effect of the blanket modules in the lower end of the frequency range (15 GHz, 18 GHz, 22.25 GHz, and 29.2 GHz) was a narrowing of the radiated beam and the appearance of side-lobes (radiation angles <90°) and back-lobes (radiation angles >90°). The beam narrowing effect had to do with the larger directivity of the radiating structure formed by the blanket surfaces and first-wall panels, into which the emission antenna radiates. The side- and back-lobes had to do with the beam truncation effect induced by the BMs’ cut-outs located near the antennas’ apertures (see Figure 14), which created reflections that, because of the broader antenna beam, were more significant at the lower frequencies. For the higher frequencies (37.3 GHz, 45 GHz, 55 GHz, 62.5 GHz, and 75 GHz), the dominant effect was the secondary lobes that appeared near the main lobe. With respect to antenna #1, the modifications induced by the blankets in the radiation patterns of antenna #2 were less noticeable, again due to the larger directivity of this antenna, which made it more insensitive to the surrounding BM structures.

The results obtained during the tests of the baseline antenna of gap 6 indicated that the BMs below and above the antennas shaped the radiation patterns and, because of the reflections and resonance effects originated in the cut-outs of the first-wall surfaces, which had a negative effect on the measurement performance below approximately 35 GHz. These effects were partially mitigated by increasing the poloidal flare of the antenna to increase its poloidal directivity. This design change significantly reduced the effects of the reflections in the first-wall cut-outs and directly improved the system’s performance.

Another critical design parameter in terms of the system performance was the gap between the antennas’ apertures and the cut-outs, in particular, at the lower end of the frequency range, where the radiation patterns are very broad. The tests have shown that if this distance was reduced from the nominal 15 mm to 10 mm the effect of the reflections in the first-wall cut-outs was reduced, improving the performance of the system.

If the above design changes were applied simultaneously, the effects of the structures surrounding the antenna were restricted to the K band (15–26.5 GHz) [20], and the measurement performance of the system was significantly improved with respect to the baseline design. Furthermore, the tests have shown that these modifications also helped in mitigating the impact of the blanket modules if, after installation, the antenna does not end up exactly centred between the blanket surfaces due to the tolerances of the different components involved.

In the case of gap 4, the in-vessel waveguides would run through the bottom of upper port 01, behind the BMs, close to the inner shell of the vacuum vessel, and down to the antennas. The waveguides would be attached to the bottom of the upper port and to the vessel’s inner shell and could not be maintained or replaced during the lifetime of ITER. The antennas, which as for gap 6 consisted of a bi-static array of parallel small-flare pyramidal horns, would be installed at the low-field side (LFS), above the midplane, in the small space between two BMs consisting of two main components: (i) the shield block, a large stainless-steel block directly attached to the vacuum vessel, whose main purpose would be to reduce the neutron flux, and (ii) the first-wall, a relatively thin structure attached to the shield block and directly facing the plasma, whose main purpose would be to sustain the high thermal and nuclear loads generated during the plasma discharges. Near the antennas, the first-wall panels are slanted both in the poloidal and toroidal directions. The 3D EM simulations have shown that the BMs would modify the antenna radiation pattern at the lower frequencies and that the geometry and location of the cut-outs required to accommodate the antennas would have a significant negative impact on the EM performance. Moreover, 2D full-wave plasma simulations confirmed the modification of the antenna radiation patterns by the blanket modules and have shown, for gap 6, that the slanted geometry of the first wall in front of the antennas could also have a strong negative effect on the measurement performance of the system. In fact, the results of the simulations indicated that there would be situations in which the system of gap 6 could not be able to meet its measurement requirements. The 2D full-wave plasma simulations also indicated potential issues with the performance of gap 4 due to the specific geometry of the enclosure, in particular, the fact that the first-wall surfaces above and below the antennas would be oblique to each other, forming an aperture that narrowed toward the plasma. Therefore, as for gap 6, it was decided to build a mock-up of gap 4 and its environment to test the performance of this system. This mock-up has been procured and provided by the ITER Organization (IO).

Figure 17 depicts the mock-up and the PPR antennas of gap 4 during the tests, which were performed with the two prototypes corresponding to the baseline and optimized antennas developed for gap 6. The results obtained with the baseline antenna have shown that the measurement performance of gap 4 was very good, with the system being able to recover all the mirror distances with position errors below the target ±1 cm margin, except in the K band, where the errors exceeded that margin. For these lower frequencies, the very broad beam radiated by the baseline antennas, due to the small poloidal flare (1 mm) was deflected and reflected in the oblique first-wall surfaces, shattering the beam and creating relevant side lobes, which produced multiple lines-of-sight that reduced the measurement performance.

With the optimized antenna, the results show that the measurement performance of gap 4 improved, mainly due to the higher poloidal directivity of the antennas, which contributed to a reduction of the level of deflections and reflections in the first-wall surfaces. Indeed, the results have shown that the performance of the system would be compliant with the target error margin across the entire operating frequency range of the PPR system (15–75 GHz), if it were not for the results obtained in the K band for the closest distance of the mirror (25 cm), where the position error was shown to exceed that margin for some frequencies, if only by a few millimeters. Finally, the results also suggested that the performance of gap 4 could be further improved to achieve full compliance by increasing the poloidal flare of the antennas, which in this gap, contrary to gap 6, would be possible due to the relatively large distance between the antennas and the surrounding components.

Concerning the measurement difficulties of gap 4 anticipated by the 2D full-wave plasma simulations, the tests with the baseline antenna confirmed, although less severe, the estimated difficulties in the lowest frequency band, K, but have not shown any problems in the highest frequency band, V. This discrepancy could be due to the 2D nature of the simulations, which, for the 3D geometry of the mock-up tests, collapses all energy in the poloidal plane and decreases the rate at which that energy dissipates, leading to an enhancement of the non-primary contributions to the measured signal [23].

### 5.5. Lessons Learned in ITER PPR

Although the ITER PPR system was descoped in 2020, the work performed until then to comply with the many interfaces and restrictions posed by its integration in ITER has revealed crucial to the development of many of the design methodologies that are currently being used in the design of a PPR system for DTT and in the conceptual design of the PPR for DEMO, both of which face similar challenges as the design for ITER. The extensive use of both EM and thermo-mechanical simulations to expedite the prototyping phase was also a key takeaway from the development of the ITER PPR system.

## 6. Microwave Reflectometry as a Plasma Position Diagnostic for DEMO

The objective of MW reflectometry in DEMO is to determine the radial edge density profile and to provide data for real-time control of plasma position and shape [26,27,28]. In order to fulfil its requirements, reflectometry shall be able to measure the plasma electron density profile in the pedestal region, where the high-density gradient occurs, with a temporal resolution of 1 ms, maximum precision error of 5%, the maximum noise level of 2%, and maximum latency of 0.01 s [29]. To be able to provide shape information, the system will use several antennas distributed at several poloidal locations, with direct access to the plasma [30,31]. Depending on the location, each reflectometer will access the plasma along a specific line of sight. Therefore, raytracing and 2D FDTD simulations were conducted to assess the expected performance of the reflectometers at the various locations. The results showed that close to the equatorial plane, a single pair of antennas can provide good spatial resolution. In regions of higher plasma curvature, however, clusters of 3–5 antennas (1 for emission and 2–4 for reception) may be needed to fully capture the reflected waves [32,33]. Considering measurements in 16 gaps on a poloidal plane around the plasma, up to 80 antennas might be required to fulfil the measurement requirements. Figure 18 shows possible poloidal locations for these antennas (100 antennas were simulated) and the simulation results for three of those locations, illustrating the need for clusters of antennas at the more extreme positions. The dimensions of the antennas and feeding waveguides are driven by the lowest frequency (15 GHz) used by the system, which corresponds to a density of $n_{e}=0.3\times 10^{19}$ $m^{-3}$.

The integration of such a high number of antennas and waveguides with the breeding blanket (BB) is by itself an extremely challenging task. In addition, these components need to endure radiation loads far exceeding those expected for ITER during long operation periods, and their design must be compatible with the DEMO Remote Maintenance (RM) systems such that they can be replaced in case of failure. The Diagnostics Slim Cassette (DSC) concept, explained in detail in Section 6.1, aims to address this requirement while providing a feasible solution for the integration of all the antennas and waveguides with the BB [29,34].

The plasma-facing antennas are foreseen to be made of tungsten, which is the material used in the BB first wall. To keep erosion below 10 μm per year, it has been recommended that the antennas are retracted from the BB surface by about 100 mm—erosion leads to higher surface roughness, which increases the reflection coefficient and the voltage standing wave ratio (VSWR), and therefore the losses. The waveguides are proposed to be made of EUROFER with copper coating for increased conductivity and must connect the antennas to the ex-vessel, crossing the primary vacuum boundary.

As with the ITER PPR system, the integration of MW reflectometry in DEMO is a multidisciplinary endeavour. EM simulations are required to optimize the location and design of the antennas, and Computer-Aided Design (CAD) and integration studies, such as space occupation, RH compatibility, and definition of interfaces with other systems, are essential. Additionally, the integration of the antennas and feeding waveguides inside the vacuum vessel will have an impact on the waveguide routing and may impose bends that, if not carefully designed, may affect the SNR. Neutronics simulations (using MCNP) [34,35] and thermo-structural analyses [36,37] play an important role not only in ensuring the reliability of the diagnostic but also in providing information about any deformations that may be induced on the components under the prescribed loads and the impact on the diagnostic measurements permitting to establish a correction to apply to minimize deformation effects if the thermal load is known.

Similar to the ITER PPR system, the additional components to implement MW reflectometry in DEMO may not be classified as requiring scheduled remote maintenance but must be prepared to an opportunistic operation of maintenance. An integrated and holistic approach is used to design the DSC considering the other systems, the expected remote maintenance operations, such as cutting/welding, alignment, grasping, manoeuvring, and transportation, the Technology Readiness Levels (TRL) of the available tools and their availability during the lifetime of the reactor. In [34] the DSC was updated to be compatible with the hybrid kinematic mechanism designed to transport the BB [38].

The maintenance addresses nominal operations, i.e., operations that are planned and expected to be performed. A failure is considered when a nominal operation is interrupted. Risk of failure may always occur no matter how well-designed and integrated the reflectometry system is in DEMO. Failure mode, effects, and criticality analyses (FMECA) are used to detect, analyse, and find solutions to avoid failures, or to mitigate their effects on the performance of the system. An initial FMECA of the DSC RH operations was performed, and mitigation actions were proposed [34]. In case of failure, the operation shall be interrupted, and the following actions may be needed: (i) start a recovery operation, where no additional equipment is required, or (ii) start a rescue operation, involving the support of additional remote maintenance systems. The recovery and rescue operations are planned to consider all possible failure situations.

### 6.1. The DSC Concept

The proposal to integrate MW reflectometry in DEMO is the Diagnostics Slim Cassette (DSC), a slim, solid EUROFER structure with 20–25 cm of toroidal thickness to be attached to or integrated with the blankets [39]. The DSC will host the antennas and their feeding waveguides, which are routed to the upper ports, as detailed in Figure 19. The plasma-facing antennas are grouped around 16 locations, known as gaps G1–G16. The gaps are planned to be arranged in three toroidal planes to avoid clashing between the waveguides (each with an inner cross-section of 19 mm × 9.5 mm).

The number of antennas in each location was derived from ray-tracing simulations [32]. However, their arrangement is also dependent on space restrictions in the upper ports, where waveguide extensions will be used to connect the DSC to the primary vacuum boundary. Due to space restrictions, the waveguides must be grouped in sockets and routed side-by-side with the blanket pipes [34]. The antenna arrangement is then influenced by two main constraints: the need to avoid clashing between waveguides belonging to different gaps and the requirement of grouping the waveguides in a small number of sockets.

To assess the possibility of symmetrical waveguide distribution, a CAD model was developed for the antenna clusters in the first wall of the DSC, aligned perpendicular to the separatrix. The options are illustrated in Figure 20 and consist of (a) two antennas aligned vertically; (b) 2 antennas aligned horizontally; (c) a cluster of 3 antennas aligned horizontally; (d) a T-shape alignment of 4 antennas, with the emitting antenna placed below the receiving antennas (suitable for the divertor region); and (e) 5 antennas, in ‘+’ and ‘×’ shapes, respectively. These two shapes, being complementary, can be used together to maximize the number of waveguides that can be routed without clashes.

The arrangement presented in Figure 20 was optimized to maximize the number of reception antennas while minimizing the number of sockets needed to route the waveguides in the upper port. It must be noticed, however, that the waveguide bends were not yet optimized from the point of view of EM performance. With all reflectometers running simultaneously in FMCW mode, the cross-interference will be challenging due to low gain horn, non-normal incident angle, wall reflection, etc. This in-vessel global microwave pollution has been clearly observed in DIII-D [40] and other facilities. Two approaches can be followed: (a) synchronized sweeping of all reflectometers with a time shift to ensure that there are no reflectometers at the same frequency at a given time; (b) Use Direct Digital Synthesis (DDS) but instead of using a linear sweep, use an encoded sweep with a different code for each reflectometer and only the received signals that show high correlation with the transmitted one will be selected. A lot of these technics are being developed for the automotive industry to overcome a similar problem of interference between the radars of different vehicles that share the same road.

One important question to address in an extreme environment such as DEMO is whether the shapes of the waveguides in the DSC are preserved to minimize power losses. Thermal analyses can be used to assess the waveguide deformation and its impact on the propagation of the EM waves. The full analysis performed for the DSC outboard segment is presented in [37,39]. The expected deformation of the waveguides in the DSC is illustrated in Figure 21, with the maximum values obtained in the antennas and waveguides located closer to the equatorial plane.

Once the expected deformations of the waveguides are known, the performance of the reflectometry system can be assessed by conducting EM analyses that include the entire transmission lines from the source up to the antennas. However, considering that the length of the in-vessel transmission lines alone is approximately 21 m (including the waveguide extensions) and the span of frequency bands, simulating the complete transmission lines would require very extensive computational resources. Therefore, critical segments of the transmission lines containing the larger expected deformations were selected for a preliminary assessment using ANSYS HFSS. The input for the simulations was the deformed geometries presented in Figure 21, divided into smaller components and then chosen for the simulations according to the deformation values. Two important waveguide sections (highlighted in Figure 21) were thus selected: a curved section just before one of the antennas and a straight section between the antennas and the upper port.

The EM analyses aimed to estimate the power losses and compare them between the undeformed and deformed waveguides, not only for the fundamental mode, TE_10_ but also for other relevant higher-order modes. Symmetry boundary conditions were applied to the waveguides to increase the simulation speed. The frequency was swept between 15 GHz and 75 GHz, with meshes optimized for each band (Ku, K, Ka, U, and V) to further optimize the simulation times. Nevertheless, all these simplifications were not sufficient within the available computational resources to complete the analysis for frequencies above 75 GHz.

Comparing the losses obtained for the two selected waveguide sections, presented in Figure 22, one can observe that the straight section exhibits, at most, a very small power loss of less than 0.12 dB, at 35 GHz. This has a minor impact on the overall system performance. On the other hand, the curved waveguide section has much higher losses, mostly for frequencies close to 50 GHz. In this case, losses above 1 dB may compromise the measurements since they will be propagated by the number of curves in each transmission line. These losses were expected, as the shape of the waveguide bends, which is critical to avoid losses due to mode conversion [41,42], is not yet optimized at this stage. Previous simulations for the 90° bends of the ITER PPR system have shown that curves with optimized hyperbolic secant shapes can eliminate losses of this magnitude [42]. These optimizations will be implemented at a later design stage when the number of antennas and waveguides and their positions within the system are fully determined.

Compared to the losses due to lack of optimization of the waveguide bends, waveguide deformation due to thermal loads has a small impact on the microwave propagation (in fact, slightly lower losses were obtained with the deformed geometries). Nevertheless, it can be concluded from this study that for optimal performance, the optimization studies must consider the operating conditions.

Multiple aspects must be considered in the design of diagnostics for DEMO, and those were taken into consideration in the design of the DSC. Space restrictions must be considered in the BBs and in the equatorial and upper ports regions. The harsh conditions of operation (radiation and heat loads) impose restrictions on the design and the selection of materials, such as in the BB region, where only metallic components are allowed, and cooling systems are needed. Additionally, plasma-facing components need to be retracted in protected locations. Finally, RH compatibility must be considered early in the design of components that will be installed in areas where hands-on maintenance will not be possible.

### 6.2. Main Achievements and Future Work on the Development of MW Reflectometry for DEMO

Fusion reactors are highly complex devices. The harsh environment and high number of interfaces between their multiple systems impose many challenges to diagnostic integration and call for an iterative design process. The MW reflectometry system for DEMO is in the conceptual design phase, and a considerable amount of work has already been accomplished concerning its integration inside the vacuum vessel. The use of high-performance codes such as REFMUL enabled the development of synthetic reflectometry diagnostics that have been used to optimize the location and distribution of diagnostic antennas. The design of the system strongly benefited from the work performed for the ITER PPR system, which has provided design solutions that have been tested in relevant prototyping activities. However, much more work is still required to address the many issues that persist:the interface between the DSC and the BB, including the respective cooling services.the detailed definition of the antenna configurations in the first wall and the waveguide routing inside the DSC.the interface for attaching/detaching the waveguide extensions to/from the BB chimneys.the limited space available in the upper port (especially at the inboard) and the need to avoid toroidal bending of the waveguides.the displacements between the vacuum vessel and the blankets, which must be accommodated by the waveguide extensions, and the design of the in-vessel/ex-vessel waveguide transitions.

The listed issues are not only dependent on the design of the reflectometry system itself but also on the outcomes of the R&D being performed for other DEMO systems, such as the BBs and the RH system. Addressing these issues requires extensive simulation work, which involves CAD design and modelling, nuclear, thermo-mechanical, seismic, EM analyses, and EM simulations. A comprehensive structural analysis will also be required to demonstrate, considering the applicable codes and standards, that the system is capable of sustaining the prescribed loads and loading conditions, thus, qualified for its intended use.

## 7. Plasma Position and Shape Reflectometry in DTT—A Test Bed for DEMO

The Divertor Test Tokamak (DTT) facility is a good test bed to test and validate relevant non-magnetic control diagnostics contributing to improving their design and implementation in DEMO. For this reason, plasma position and shape reflectometry (PP&SR) is being investigated for installation on DTT with the aim of testing its capabilities in view of its application in DEMO as a complementary/backup system for magnetic sensors. In DTT, the PP&SR system should use four lines of sight: one in the LFS midplane, one in the upper vertical port, and two in the HFS region. Depending on the physical space available, a full band configuration (K, Ka, U, V, W, and D band) or only a subset of it will be used by the reflectometers at each line of sight to monitor the full radial density profile or the regions around the last closed flux surface. The utilization of FDTD codes enables a detailed description of the reflectometers, which includes aspects such as propagation in realistic plasmas, the specific location within the vacuum vessel, and access to the plasma. Additionally, the results of these simulations can be used to test new signal-processing techniques. To predict the behaviour and capabilities of the proposed reflectometry system for DTT, three synthetic diagnostics (SD) were prepared. These SDs correspond to three O-mode reflectometers located on the LFS, with one on the midplane and two others located away from the midplane [43]. The simulations were performed using the 2D FDTD full-wave code REFMULF in two plasma scenarios envisioned for DTT, namely the 5 MA single null (SN) and double null (DN) configurations during the start-of-flat (SOF) phase of the discharge. One of these simulated SD, located at the upper section of the LFS, is depicted in Figure 23, showing its position on a cross-section of the DTT vacuum vessel. This is an example of non-standard PPR, which may play a key role in next-generation machines. Figure 24 depicts snapshots of the full system 2D simulations for this gap during an SN plasma at four frequencies corresponding to the K (f=26.5 GHz), Ka (f=40 GHz), V (f=75 GHz), and W (f=95 GHz) bands. Note that the snapshot for the K band is the one represented in machine coordinates in Figure 23. Although it is not shown here, the Q band was also simulated. A simulated FMCW signal is excited at the antenna with frequencies ranging from the minimum to the maximum frequency of each band. The probing beam propagates to the plasma and is reflected back to the antennas, where it is decoupled from the emission through the use of a Unidirectional Transparent Source (UTS) [44]. These simulations include the plasma curvature and the full structure surrounding the plasma resulting in a visible distortion of the field patterns with the appearance of resonances in the lower K and Ka bands passing to a regime of multi-reflections at higher frequency bands. Ancillary simulations performed with slab plasmas were used to obtain reference cases against which the performance of the different gaps is evaluated, in particular, the measurement error. Results from the simulations of reflectometers at LFS, in equatorial, upper, and lower sections of the machine demonstrate good performance with measurement errors within the required ±1 cm and small deviations, of the order of some millimeters, at the separatrix. These simulations consider the plasma and surrounding structures, as well as their interplay, demonstrating that the proposed reflectometer positions (or any small variations) are suitable for DTT. Since 3D simulations are computationally intensive and require access to high-performance computer (HPC) facilities, the use of 2D codes is more common. However, it is important to evaluate if a 2D model is sufficient to describe the problem at hand and what compromises may be involved in using such a model or if a 3D approach is necessary, especially since the received power can be a concern on larger devices. To address this, simulations were carried out using two full-wave codes, namely REFMULF (2D) and REFMUL3 (3D), to make a comparison between the two [45].

The overall results are encouraging for the implementation of reflectometry systems at DTT and represent an important milestone for reflectometry in D-shaped tokamaks. Even a reduced system with an equatorial gap and only one of the other two gaps (either on the upper or lower part of the machine) would be a significant advancement for reflectometry in general and for DTT and DEMO, as reflectometry has never been used before to measure the density profile from a location away from the midplane in a D-shaped tokamak.

## 8. Development of a Compact Reflectometer

The number of channels needed to achieve good poloidal and toroidal plasma coverage in fusion reactors such as DEMO is larger than in current machines, which could take advantage of the use of compact and modular millimetre wave back-ends. For that reason, Instituto de Plasmas e Fusão Nuclear at Instituto Superior Técnico (IPFN-IST) in Lisbon is developing and prototyping a coherent fast frequency sweeping RF back-end using commercial Monolithic Microwave Integrated Circuits (MMICs) leveraging the large-scale availability of high-performance MMIC at affordable prices.

The new back-end system was designed to be able to accommodate different frequency ranges in a flexible way. Thus, the core back-end covers the NATO J-band (10 GHz to 20 GHz) and is capable of driving external full-band frequency multipliers to achieve ultra-wideband coverage up to 160 GHz. Although developed for fusion plasma diagnostics, this new compact reflectometer may also be used in other FMCW radar applications, such as space re-entry experiments. The details of its implementation are described in [46].

The system architecture is shown in Figure 25. Heterodyne radars use a microwave sweep oscillator to generate a signal, which is then used for different purposes in the circuitry: (i) for target probing through the transmission channel (Ax); (ii) to drive a frequency translator. The frequency-translated signal is used to drive two mixers: the front-end mixer of the receiver channel (Bx), which collects the reflecting signal from the target and generates the beat signal around the Intermediate Frequency (IF) carrier, and the local reference mixer that generates the reference signal used on the quadrature detector to demodulate the reflected signal. The fact that the IF is generated by the frequency translation of the probing signal ensures the coherence of the system [47]. This approach improves the radar dynamic range with respect to homodyne detection. The developed system uses two high-frequency PCB boards: (i) the back-end board, which is responsible for generating all the signals (shaded green in Figure 25), and the quadrature detector board, which is used to demodulate the reflected signal, providing amplitude and phase information (shaded pink in Figure 25). The front-end part of the system (yellow-shaded area in Figure 25) includes the final frequency multipliers, mixers, transmission lines, and antennas. The compact reflectometer will have to face two major challenges, (i) high performance with excellent shielding against X-ray, Gamma, and neutron; (ii) affordable price. The prototype was developed with commercial circuit components on a customized PCB base. It is one possible solution, but some may consider it outdated. The most promising solution for high-level integration microwave diagnostics is system-on-chip, which develops all microwave circuits on a $10\;{\mathrm{mm}}^{2}$ semi-conductor chip (CMOS, InP, GaAs, and GaN). This technology has been successfully developed on DIII-D since 2019 [48]. However, the reflectometers on a chip on DIII-D were applied to imaging reflectometry where fixed frequency or very limited frequency sweep is used. The presented prototype could have been produced with a larger level of integration, but in this way, it was possible to better optimize the different stages of the reflectometer. Radar on a chip exists for massive applications in the automotive industry but is bandwidth limited. It should also be noted that the US Domestic Agency, which is responsible for the development of the main plasma low field side ITER reflectometers, did not propose the “on a chip” solution for those systems. However, it is expectable that the future will lead to a large integration system on a chip or a couple of chips, and those advances will be considered for future developments as they will require a larger number of reflectometers.

Figure 26 shows a prototype of the compact reflectometer being tested. To be capable of efficiently driving external frequency multipliers, the system generates full band signals exceeding 8 dBm. To keep all undesirable harmonics at least 15 dB below the desired output across the entire frequency range of interest, all signal paths were fine-tuned using appropriate in-line attenuators. The system also proved capable of coping with low-level reflected signals.

The back-end and in-phase and quadrature (I/Q) boards included EM shielding and were designed for easy installation on a standard 3U 19′′ rack.

The precision of FMCW radars is known to be strongly dependent on the linearity of the frequency sweep. Modern fully integrated Direct Digital Synthesis (DDS) chips can generate signals with very high frequency, phase precision, and resolution, improving the linearity of the radar probing signals and allowing very linear and agile frequency chirps. The direct digitalization of the IF signals makes it possible to use highly flexible data processing techniques. A new prototype using these two techniques is being developed. The system is expected to be ready for deployment into experimental devices by early 2024.

## 9. Conclusions

In 2017, the US National Academy of Engineering considered that providing energy from fusion was one of the grand challenges for engineering. The engineering community faces the challenge of scaling up the fusion process to commercial proportions in an efficient, cost-effective, and environmentally friendly manner. Controlling the burning plasma is one of the critical issues that need to be addressed. Plasma Position Reflectometry is expected to have an important role in next-generation fusion machines, such as DEMO, as a diagnostic to monitor the position and shape of the plasma. Proofs of concept focused on the ability to use the diagnostic for plasma control were performed in the ASDEX-Upgrade and COMPASS tokamaks. The success of these experiments has shown that it is possible to fulfil the ITER and DEMO requirements for plasma control using microwave reflectometry as an alternative to standard magnetic measurements. Namely, it can achieve the high availability and reliability of plasma control required for the operation of ITER and DEMO, helping to pave the way towards the realization of future fusion power plants.

Due to the nature of microwave reflectometry measurements, the front end of the reflectometers (antennas and feeding waveguides) is directly exposed to the plasma, thus subjected to fluxes of high-energy neutrons (14 MeV) and high thermal loads. Therefore, it is indispensable to ensure that these components can operate in the harsh environment of a fusion reactor, where they will face fusion powers of 500 MW for ITER and 2000 MW for DEMO (heat fluxes up to 1–2 MW/$m^{2}$ [49]). For that purpose, complex design studies (involving neutronics, thermo-mechanical studies, and EM analyses) are crucial to validate the design of these components and to ensure that they can sustain the expected loads without damages that could seriously compromise their performance and low aging to ensure their time life to be compatible with a fusion power plant. The neutronics and thermo-mechanical studies provide an insight into what needs to be optimized in the design to meet certain requirements, while the EM studies provide an understanding of the impact of the thermo-mechanical loads on the ability of the systems to perform measurements.

Some of the developments achieved so far have strongly benefited from the progress made for the REFMUL suite of codes developed at IPFN for reflectometry simulation. The suite includes REFMUL, a 2D O-mode simulation code, and REFMULX/REFMULXp, a 2D X-mode (serial/parallel) simulation code. Recently, two new codes were added: REFMULF and REFMUL3. REFMULF is a 2D code able to cope with full polarisation waves, treating all components of the electric and magnetic fields of the wave [50]. REFMUL3, which recently entered production, is a 3D full-wave code. Additional tools were developed to automate the simulation and data analysis of a generic PPR system and make it adaptable to different devices. These techniques can be used to optimize a given PPR system for a given equilibrium scenario and study its stability. Furthermore, an effort has been devoted to EM modelling to further understand how edge turbulence impacts density profile reconstruction using O-mode reflectometry [51].

The ITER PPR system would allow testing the use of this diagnostic for control of burning plasma, assess its performance under harsh conditions, and evaluate the design decisions taken and how those would affect the performance of the diagnostic. The R&D and design of the in-vessel components progressed according to schedule. Unfortunately, ITER terminated the PPR project due to concerns that captive components (waveguides and supports needed to install the system between the ports and the diagnostic hall) would not be procured and delivered in time. This would have resulted in unacceptable delays in installing the remaining captive components from other systems that could only be installed after the PPR system components were fully in place. As a result, ITER opted to rely on alternative technologies for measuring plasma position. Although descoped, the design of the ITER PPR system allowed the development and testing of some important aspects of using reflectometry as a plasma position system and the identification of critical issues that still require further development or prototyping.

The MW reflectometry system being proposed for DEMO is at an early stage of development, and a significant amount of work is still required. The work performed so far illustrates the level of multidisciplinary required to develop and integrate microwave reflectometry diagnostics on future fusion devices such as DTT and DEMO. Such developments require an integrated approach that encompasses, for instance, microwave reflectometry simulations, CAD design and integration studies (space occupation, RH compatibility, definition of interfaces with other systems, etc.), neutronics simulations (MCNP), thermo-mechanical analyses, EM full-wave simulations and multiple design iterations, incorporating constraints from each area.

Many of the developed concepts will have to be further tested in the laboratory and in fusion experiments. DTT will provide a unique opportunity to test further the use of reflectometry for plasma control with the aim of testing its capabilities in view of a possible reactor application as a supplementary/backup system for the magnetics.

The recent developments in the design of a compact reflectometer will also be crucial for the successful integration of many measurement channels in the reduced space available in future fusion machines.

## Figures and Tables

**Figure 1 sensors-23-03926-f001:**
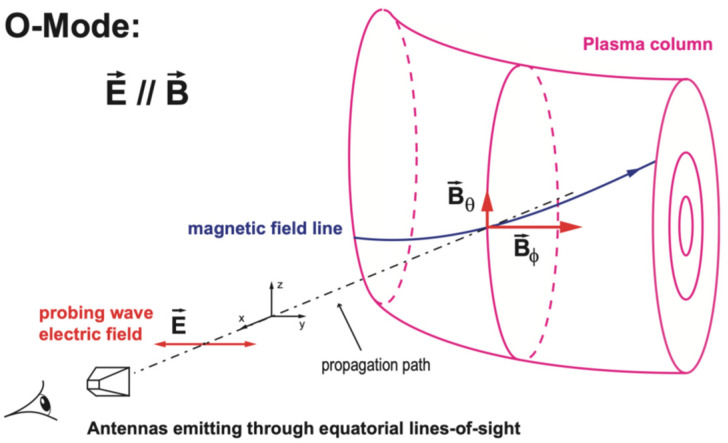
Schematic view of the O-mode reflectometry set-up where the wave electric field is not perfectly aligned with the tokamak magnetic field.

**Figure 2 sensors-23-03926-f002:**
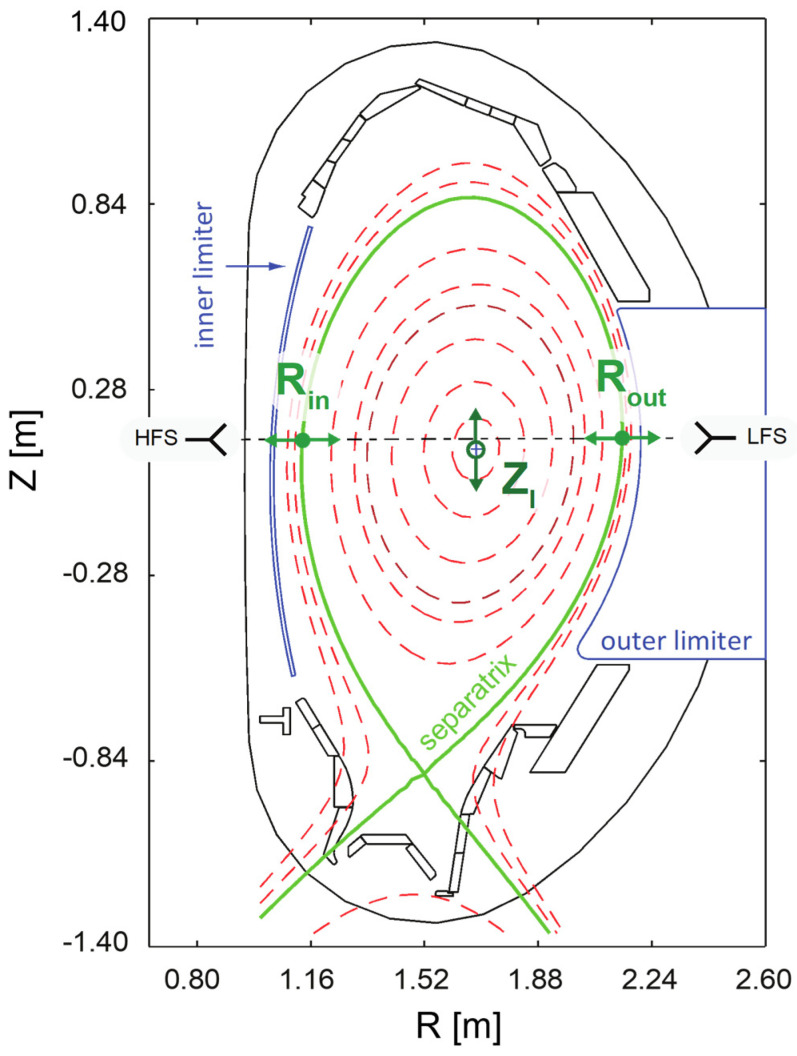
Simplified view of an ASDEX Upgrade poloidal cross-section, showing the magnetic separatrix position, the points used ($R_{in}$, $R_{out}$, $Z_{I}$) for the plasma position, and the lines-of-sight of the HFS and LFS microwave reflectometers.

**Figure 3 sensors-23-03926-f003:**
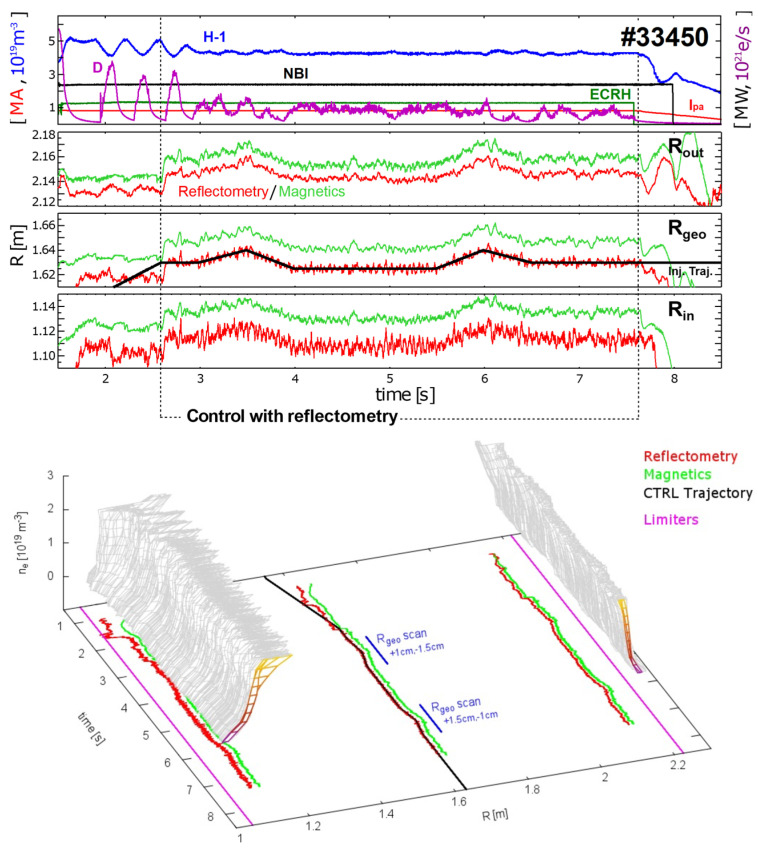
Time traces during ASDEX Upgrade discharge #33450 showing the position controller target trajectory and the magnetic and reflectometric separatrix positions (the reflectometry-based control of the plasma was performed during the highlighted period, ∼2.6–7.6 s).

**Figure 4 sensors-23-03926-f004:**
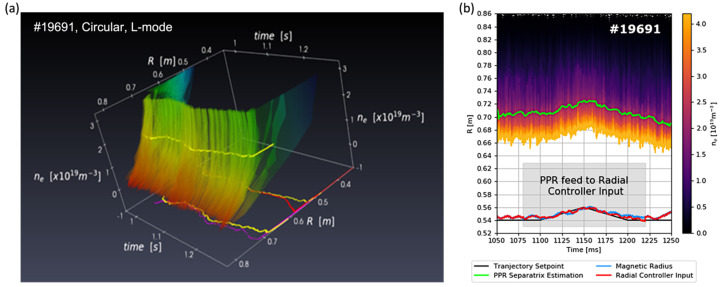
(**a**) Real-time reflectometry profiles obtained during COMPASS discharge #19691; and (**b**) estimation of the separatrix location based on reflectometry profiles (green) and controller input magnetic radius (red). In the shaded region, the controller input was the magnetic radius derived from the reflectometry separatrix position estimation and not the actual magnetic measurement (blue).

**Figure 5 sensors-23-03926-f005:**
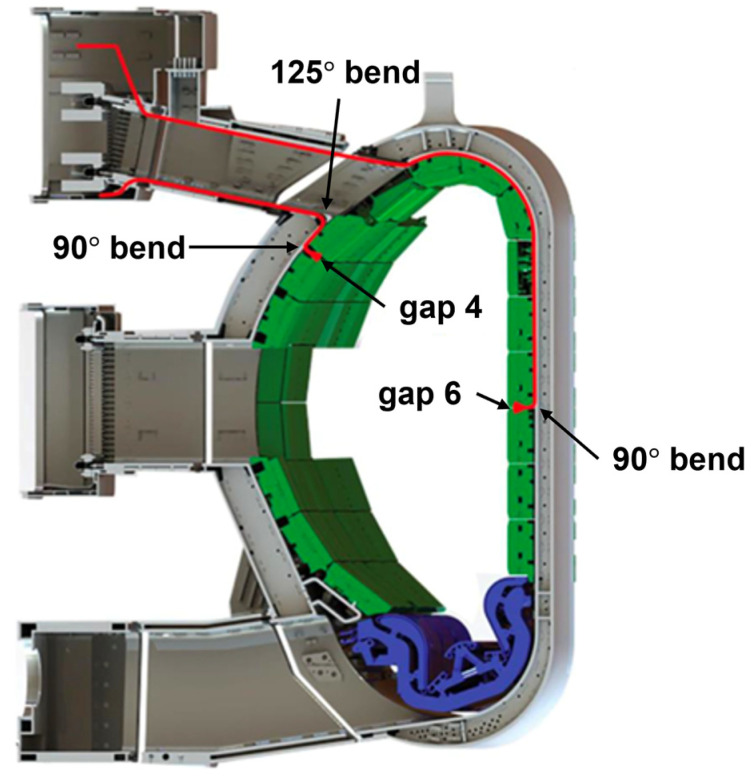
Location (red) of the ITER PPR in-vessel components of gaps 4 and 6, indicating some of the parts that required a detailed design, namely the 125° bend, the 90° bends, and the antennas. The in-vessel waveguides are routed behind the blanket modules (coloured green) near the vacuum vessel wall. The divertor structure is represented in blue.

**Figure 6 sensors-23-03926-f006:**
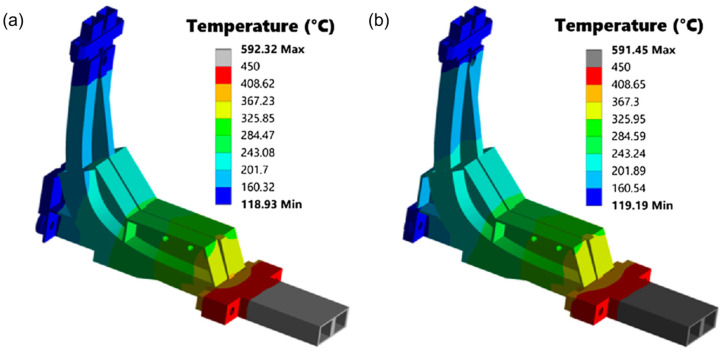
Temperature distributions of the front-end components of gap 6 after extending the design volume in the poloidal direction (**a**) and in the toroidal direction (**b**).

**Figure 7 sensors-23-03926-f007:**
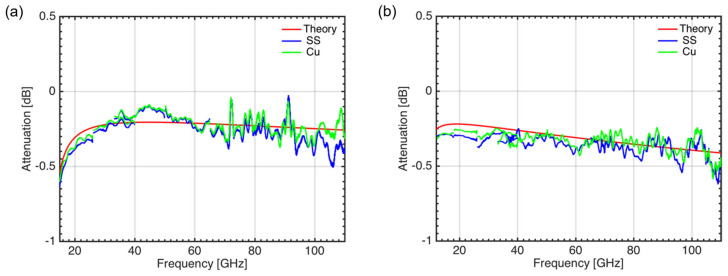
Performance of the stainless-steel copper-coated waveguides (blue) against the copper-only waveguides (green) and theoretical predictions (red) both for O-mode (**a**) and X-mode- (**b**) like polarizations.

**Figure 8 sensors-23-03926-f008:**
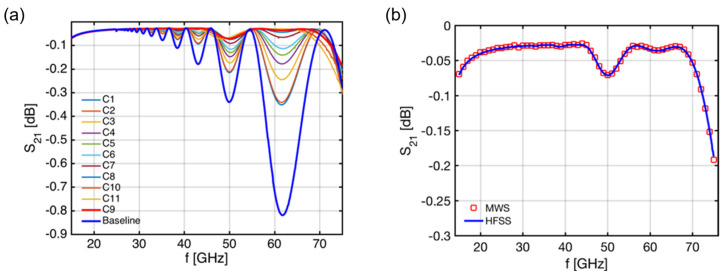
Performance of the constant-radius bend (baseline design) and of various hyperbolic secant 125° bends (C1 to C9) (**a**) and comparison between the results obtained with HFSS and MWS for the optimized design (C9) (**b**).

**Figure 9 sensors-23-03926-f009:**
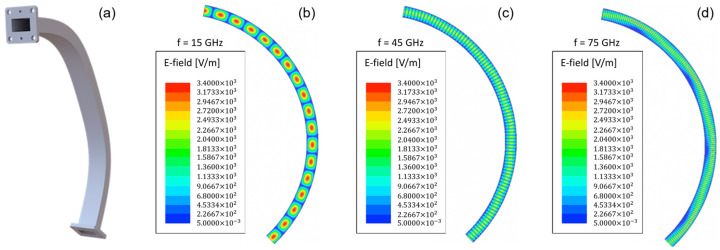
Electric field distribution inside the optimized (C9) 125° bend (**a**) obtained with HFSS for (**b**) 15 GHz, (**c**) 45 GHz, and (**d**) 75 GHz.

**Figure 10 sensors-23-03926-f010:**
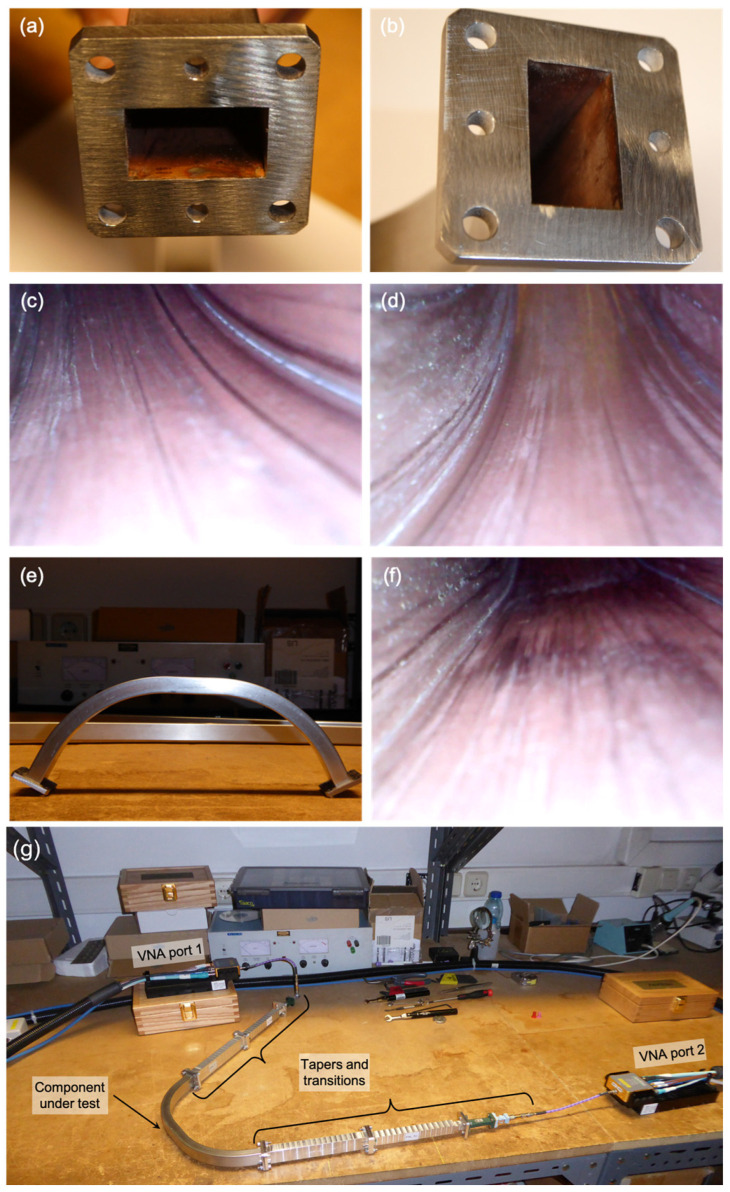
Prototype of the 125° bend (**a**–**f**) and experimental setup at IPFN-IST microwave laboratory (**g**) used to measure the EM performance of the prototype samples.

**Figure 11 sensors-23-03926-f011:**
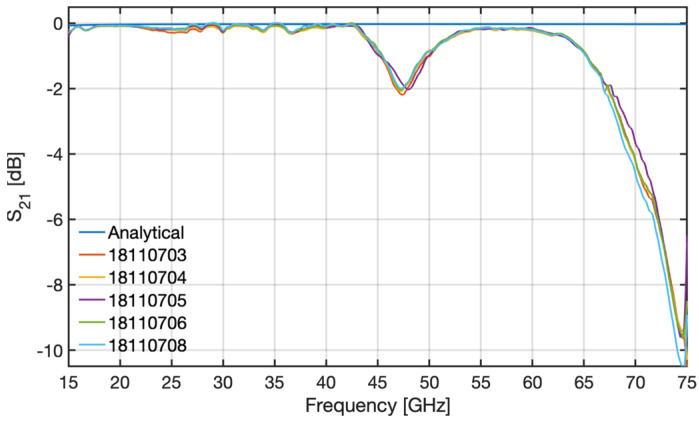
Analytical losses and experimental attenuation for the various prototype samples (numbered 18110703 to 18110706 and 18110708) of the 125° hyperbolic secant rectangular (20×12 mm) stainless-steel copper-coated waveguide bend. The analytical losses were calculated using Equation (3) for a straight rectangular waveguide with the same dimensions and equivalent length of 0.33 m.

**Figure 12 sensors-23-03926-f012:**
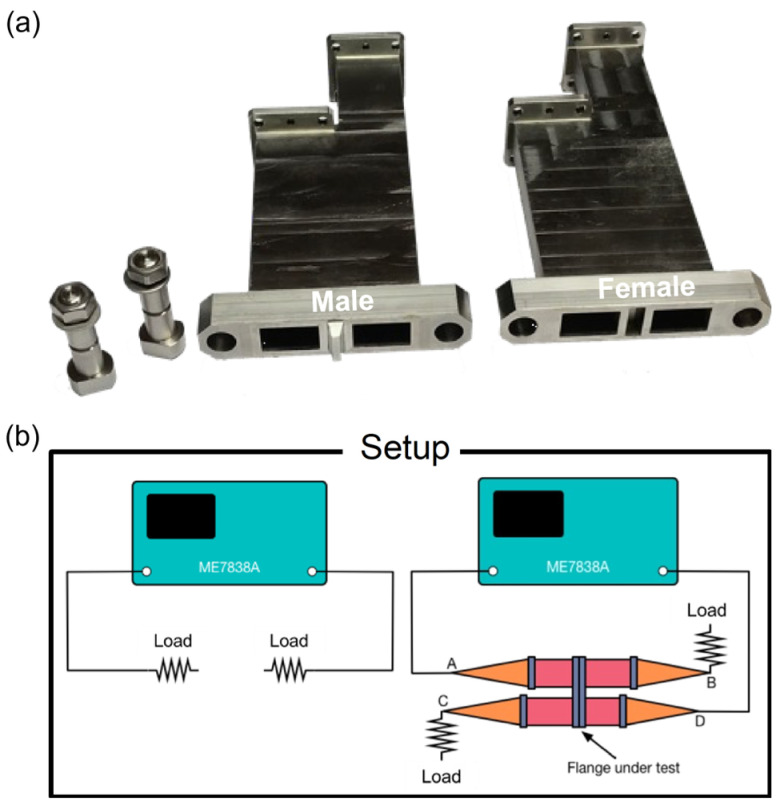
Prototype flanges designed to avoid crosstalk (**a**) and experimental setup to measure the crosstalk at the flanges (**b**).

**Figure 13 sensors-23-03926-f013:**
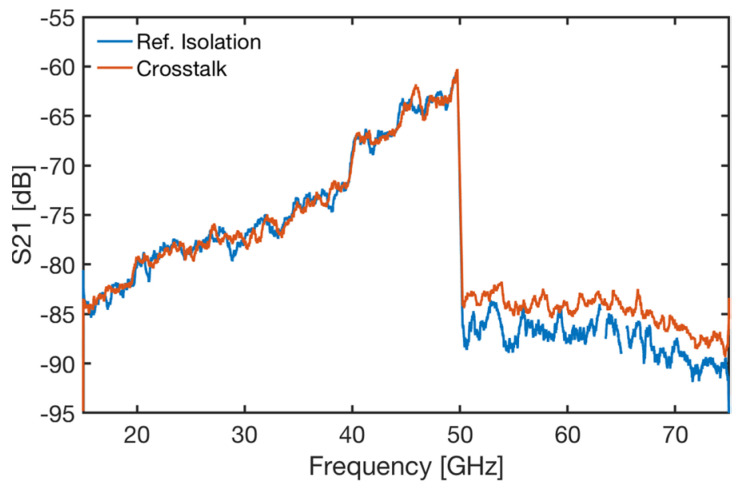
Crosstalk measurements on the prototype flange versus frequency. The discontinuity at 50 GHz is due to the different sensitivity of the VNA in the 50–75 GHz band.

**Figure 14 sensors-23-03926-f014:**
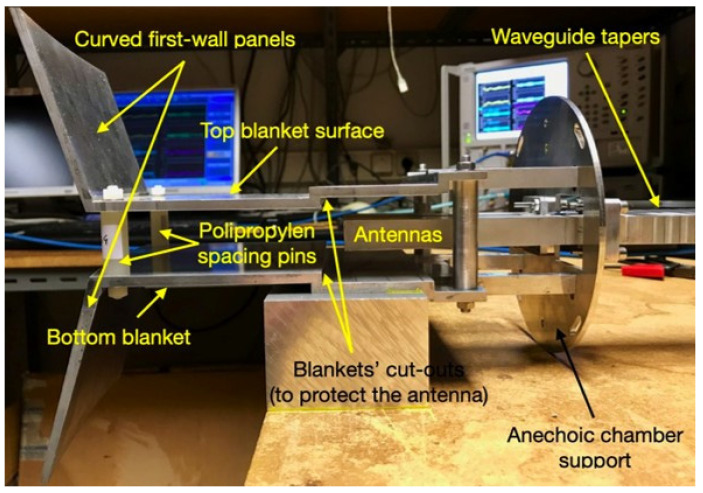
Prototype antennas and mock-up of the BMs for gap 6.

**Figure 15 sensors-23-03926-f015:**
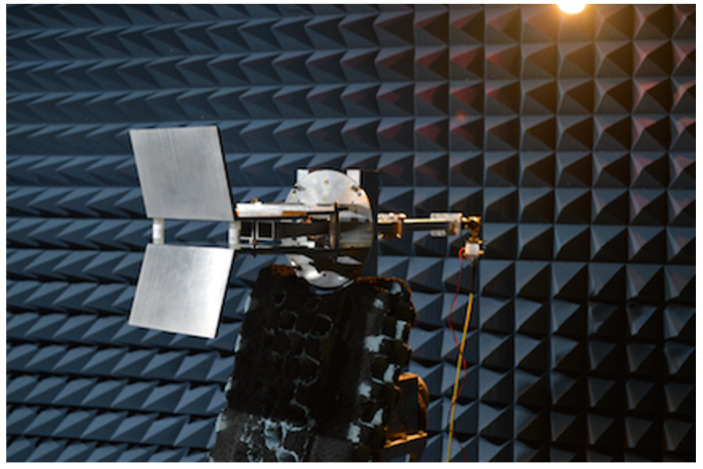
Prototype antenna assembly for gap 6 inside the anechoic chamber during the measurement of the radiation patterns.

**Figure 16 sensors-23-03926-f016:**
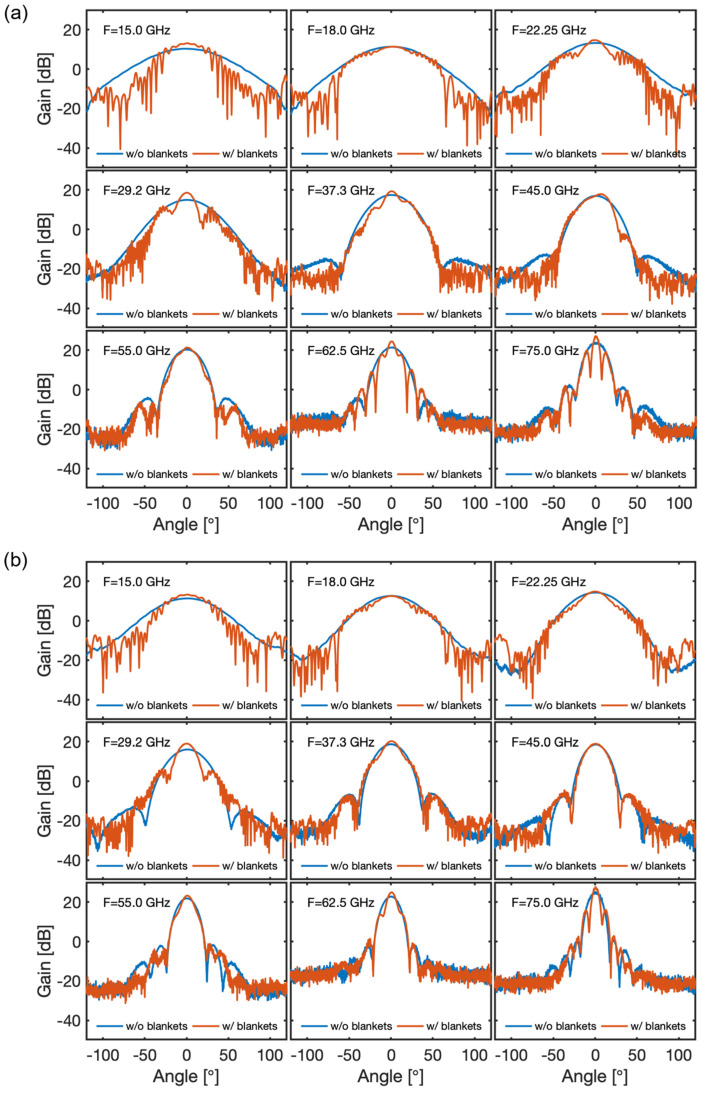
H-plane co-polar radiation patterns for antenna #1 (**a**) and antenna #2 (**b**), with (red) and without (blue) the BMs.

**Figure 17 sensors-23-03926-f017:**
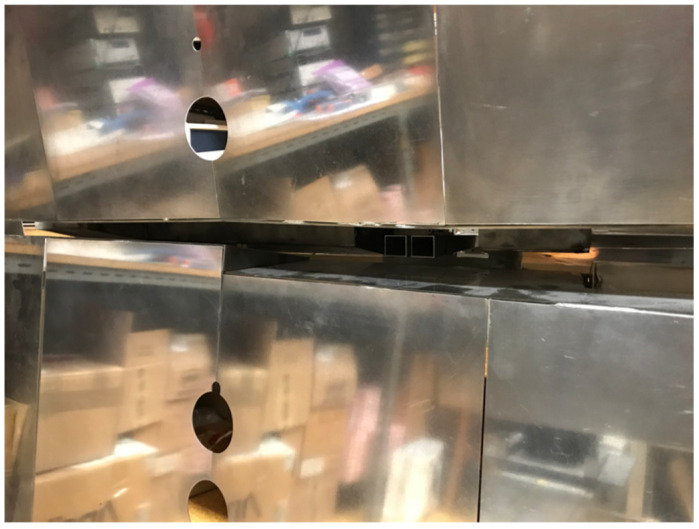
Mock-up of gap 4 provided by the IO featuring the relevant geometry of the first-wall surfaces and BMs surrounding the PPR antennas.

**Figure 18 sensors-23-03926-f018:**
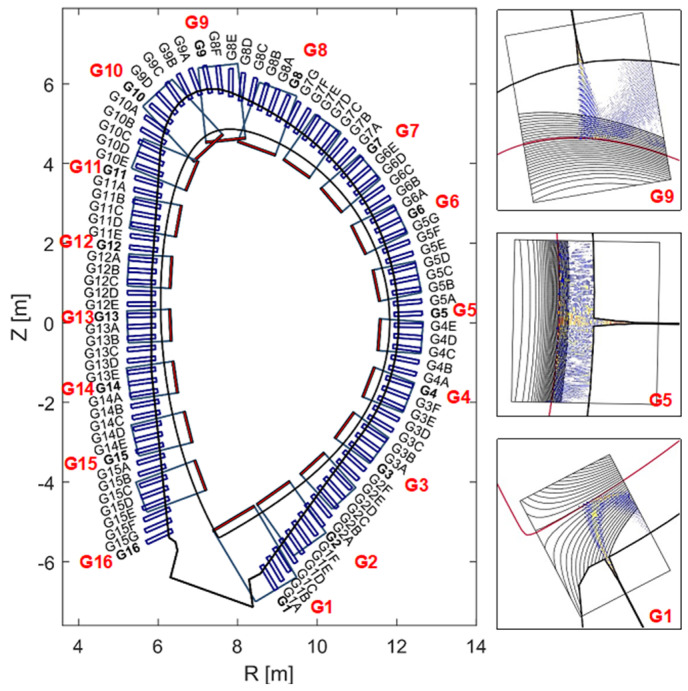
Poloidal distribution of the antennas aligned perpendicularly to the separatrix tangent line (**left**) and their expected performance at three locations (**right**) for DEMO when the LOS is set perpendicular to the wall. The smaller blue boxes correspond to the region to place the antenna setup, and the larger blue rectangles correspond to some examples of the selected region of interest for the tested poloidal positions [31].

**Figure 19 sensors-23-03926-f019:**
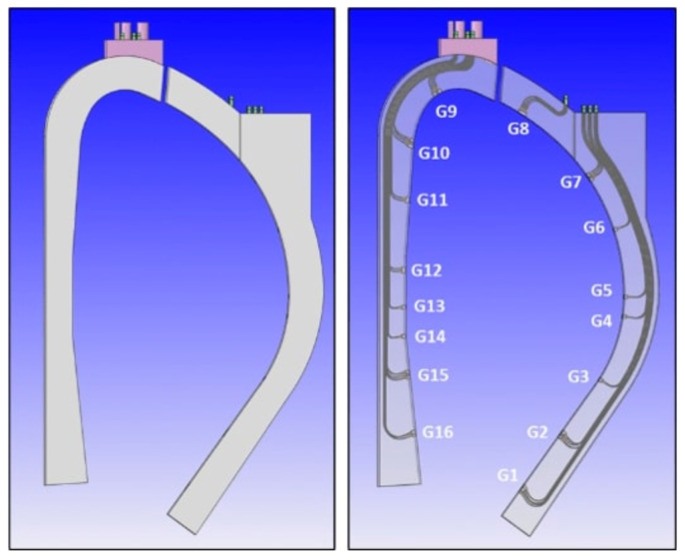
Poloidal view of the DSC with (**left**) and without (**right**) applying transparency showing the waveguide routing.

**Figure 20 sensors-23-03926-f020:**
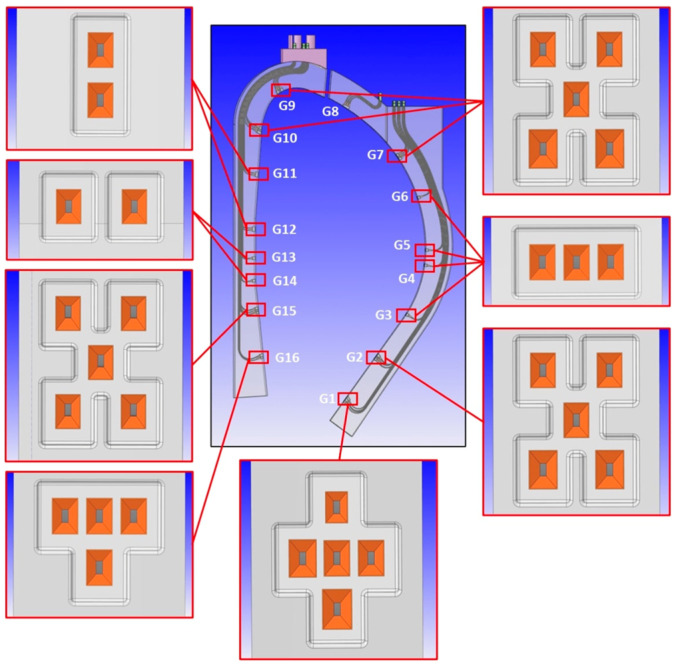
Possible distribution of antennas in the first-wall DSC openings [37].

**Figure 21 sensors-23-03926-f021:**
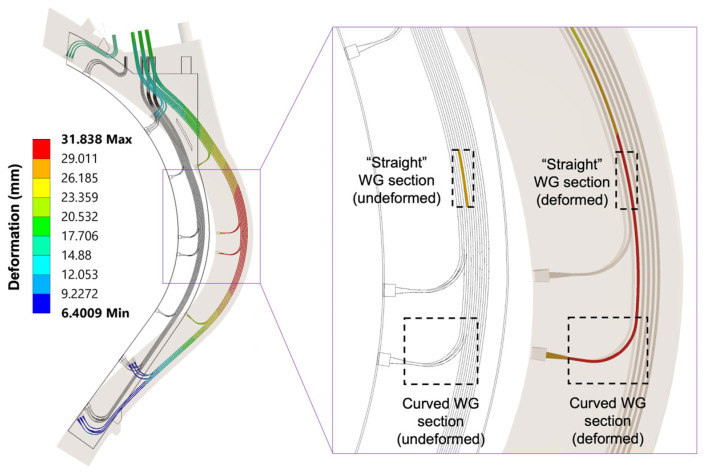
Preliminary results for the expected deformation of the waveguides inside the DSC (exaggerated representation). The waveguide sections selected for the preliminary EM performance assessment are highlighted on the inset.

**Figure 22 sensors-23-03926-f022:**
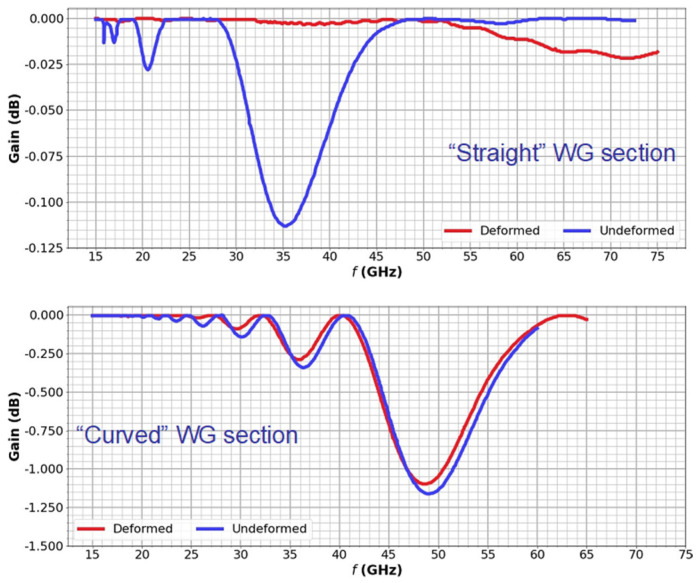
Attenuation of the TE_10_ mode for the two selected waveguide sections.

**Figure 23 sensors-23-03926-f023:**
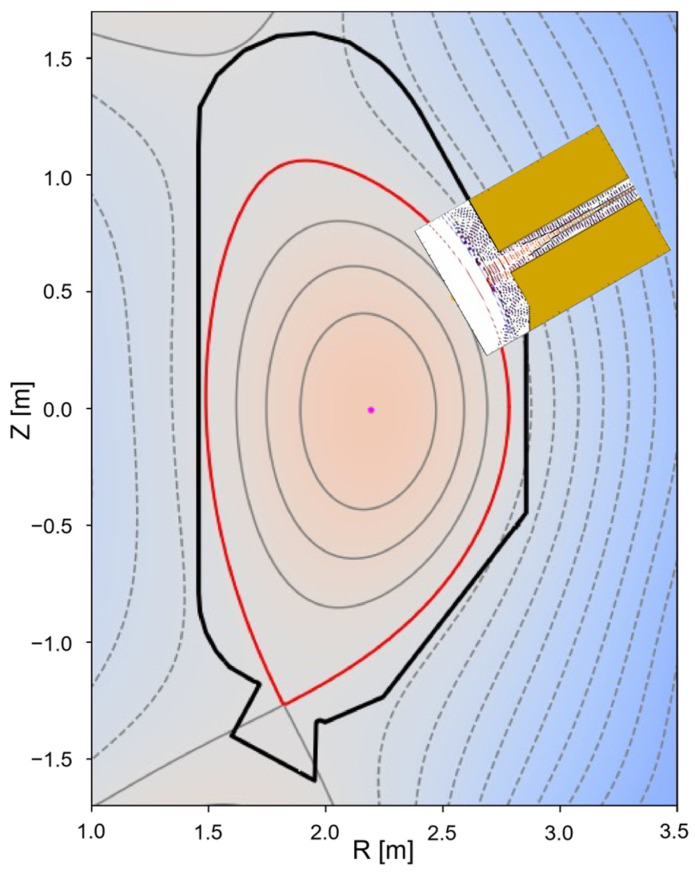
Cross-section of the DTT vacuum vessel showing an LFS synthetic reflectometer placed at the upper part of the machine. The line in red represents the Last Closed Flux Surface (LCFS).

**Figure 24 sensors-23-03926-f024:**
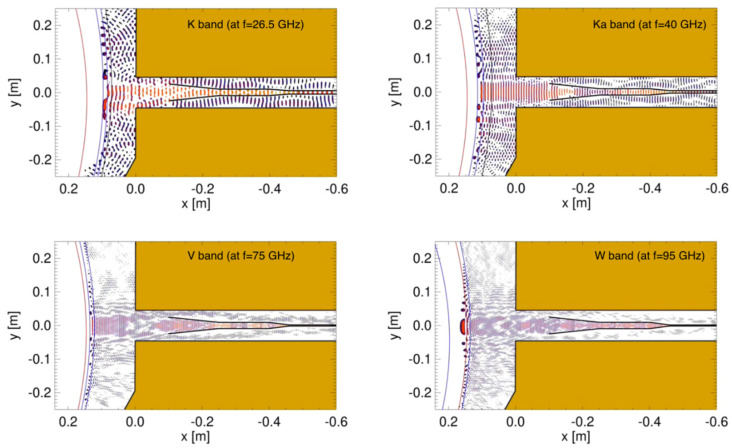
Snapshot of electric field $E_{z}$ for LFS SD located at the upper part of the machine (see Figure 23) in the DTT f=26.5 GHz (top left), Ka band at f=40 GHz (top right), V band at f=75 GHz (bottom left) and W band at *f* = 95 GHz (bottom right). The vertical blue lines mark the iso-density region corresponding to the limits of the band; the red line shows the position of the separatrix.

**Figure 25 sensors-23-03926-f025:**
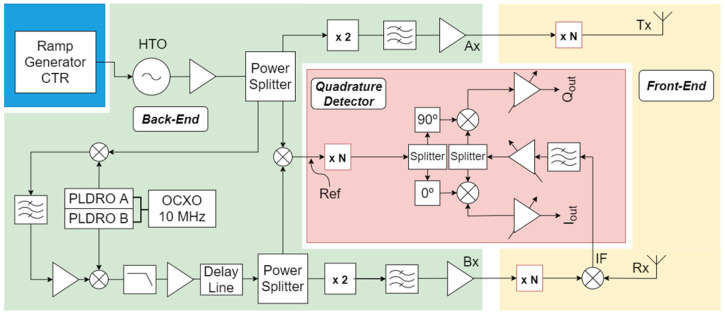
Block diagram of the compact reflectometer.

**Figure 26 sensors-23-03926-f026:**
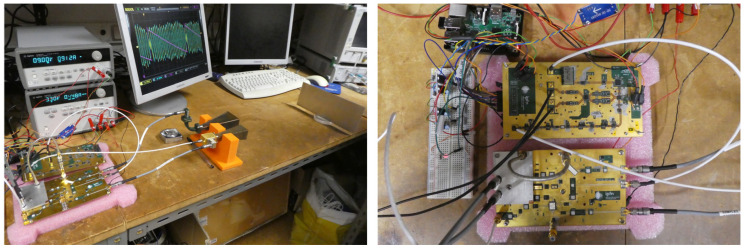
Compact reflectometer prototype being tested with a metallic mirror, setup (**left**), and top view on the electronic hardware (**right**).

## Data Availability

Not applicable.

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
