# Peer review of "Advances, Challenges, and Future Perspectives of Microwave Reflectometry for Plasma Position and Shape Control on Future Nuclear Fusion Devices"

_sensors, 2023, doi:10.3390/s23083926_

Round 1
Reviewer 1 Report
The manuscript is in the scope of the journal. It concerns the perspectives of using microwave reflectometry to control plasmas in future fusion-based devices like DEMO. Such diagnostics system is not foreseen to replace magnetic diagnostics but in complementary to them. The authors mention that such a diagnstic system will not be installed on ITER even though it was considered during the phase design until recently. However, the authors did not explain why it has been abandoned for ITER.
Generally speaking, it is interesting to summarize the research on this diagnostic system from both the experimental and numerical aspects. The manuscript is rather long and I think it must be shortened. In particular, there are some figures, which in my opinion, are not necessary and can be removed. Another concern is the large number of abbreviations which are used and this may prevent non specialist readers from understanding the main points and results of this manuscript. I recommend to use full terms instead of abbreviations when these term do not appear too much. Also may be a golossary will be useful. There are also approximations in terms of terminology for instance the authors use the term plasma column when speaking about tokamak plasmas which have generally a donut shape. In the following you find a non-exhaustive list of comments page per page.
-Page 2.
*L58-59. A temperature hotter than the core of the Sun: I would say a temperature higher than that of the core of the Sun. The term column is not appropriate for tokamak plasmas. It must be replaced or removed.
-Page 3.
*L111. Add "of the magnetic field" to "poloidal component"
-Page 4
*L157-158. Add of between "use" and "microwave". To control of the plasma--> To control the plasma..
-Page 5
*Figure 2. cross-section--> poloidal cross-section. At the end of the straight line showing the separatrix there is a small cicrle which may lead to a misunderstanding, the separatrix being the whole green line.
*L205. The term plasma column is used. Is it for a cylindrical plasma?
-Page 6
*L223-224. Add the term "power" to NBI. The term ELMy is not explained (Edge Localized Modes), few words about this instability may be useful.
* In the top panel of figure 3, $R_{aus}$ appears instead of $R_{out}$.
-Page 7
*L249. Add "system" to PPR
*L281. What does mean a control loop?
-Page 8
*Figure 4. The grey zone extends over almost 150 ms according to the time axis unit. It is not easy to understand the link bteween this zone and the control loop of 500 microns (0.5 ms). This is about 1/300 with respect of the duration of the grey zone. An explanation will be helpful.
* Section 5. There is an issue with the use of the past. You are talking about events and things planned in the past for future even though they have been abandoned or amended.
-Page 9.
*Figure 6. Is it useful?
-Page 10.
Figure 8. Is it useful? The shown information can be explained with some words in the text.
-Page 12.
* figure 10. What do C_1, C_2, C_3 stand for?
* MWS and HFSS. What are the main differences between these methods? How accurate are their results?
-Page 14
*Figure 13. what do numbers stand for?
-Page 15.
*Figure 14. Top panel, you may add male and femae terms. The bottom panel is not clear for me.
*Figure 15. What is the explanation of the discontinuity around 50 GHz?
*L470. FTDT-->FDTD
-Page 16.
* Figures 16 and 17. Are they necessary? Both of them?
-Page 17.
*L524-527. A bit complicated to understand. Higher frequencies ? which ones?
-Page 18.
*L528-535. It is better to indicate on the figures these lobes.
-Page 19.
IO-->Iter Organization (IO)
-Page 20
*L623 [21][22][23]-->[21-23]
*L626. maximum precision of 5%? (minimum precision 5% instead no?)
-Page 21.
*L655. [23][27]-->[23,27]
*L665. CAD?
-Page 26
* Figure 26. There is a missing interpretation of the shown pictures.
-Pages 27-28
*Ref [38] is not cited.
*Figure 28. Is this figure important to keep?
*L912. Where is and who is developping this new prototype? When would it be ready (in few months?).
-Page 29.
*Ref [42] is cited before ref [41]. To permute.
-L954. What the ITER PPR system was descoped? Some words would be very useful to understand. Now you are talking about DEMO.
Author Response
We would like to thank the reviewers for the comments that helped to improve the readability of the paper. Below you can find a detailed reply to the reviewers regarding the changes introduced.
Upon your suggestions we removed some figures. In some cases the relevant information was added to the other figures or described in the text. Minor rearrangements of the figures were done to make better use of the space available.
Page 2
*L58-59 - A temperature hotter than the core of the Sun: I would say a temperature higher than that of the core of the Sun. The term column is not appropriate for tokamak plasmas. It must be replaced or removed.
The sentence was corrected, and the term “hot plasma column” replaced by “hot plasma volume”.
Page 3
*L111 - Add "of the magnetic field" to "poloidal component".
Done.
Page 4
*L157-158 - Add of between "use" and "microwave". To control of the plasma--> To control the plasma.
Done.
Page 5
*Figure 2 cross-section --> poloidal cross-section. At the end of the straight line showing the separatrix there is a small circle which may lead to a misunderstanding, the separatrix being the whole green line.
The term “poloidal” was added to the caption. The figure was revised to place the indication of the separatrix along the green line to make it clearer
*L205. The term plasma column is used. Is it for a cylindrical plasma?
The term plasma column has been historically used in tokamak plasmas although we are not referring to cylindrical plasmas, e.g. https://iopscience.iop.org/article/10.1088/0029-5515/23/10/005, however we acknowledge the correction and replaced column by volume.
Page 6
*L223-224. Add the term "power" to NBI. The term ELMy is not explained (Edge Localized Modes), few words about this instability may be useful.
Added the term “power”.
Added the sentence “The experiment was performed during the high-confinement mode (‘H-mode’) in the presence of Edge Localized Modes (ELMs) instabilities which results in bursts of energy and particles at the plasma edge. Electron density profiles measured by reflectometry can be severely affected by the ELMs and was required to develop algorithms to automatically validate reflectometry measurements to cope with the effects of these cyclic transitory periods.”
*In the top panel of figure 3, $R_{aus}$ appears instead of $R_{out}$.
Thank you for spotting the incorrection . Raus was replaced by Rout.
Page 7
*L249. Add "system" to PPR.
Done
*L281. What does mean a control loop?
It was corrected and better clarified in the text. In this context it was meant control-loop cycle time and not control-loop per se. the text now is written as “Figure 4.a) shows the real-time reflectometry profiles for PPR control demonstration discharge 19691 (circular L-mode discharge). Figure 4.b) shows the separatrix location estimated from the reflectometry profiles (green) from which the PPR magnetic radius controller input was derived. During the period highlighted in grey, ~140 ms, the estimated magnetic radius was used for the radial position controller input (red) replacing the actual magnetic measurement (blue) in the slow controller (500 μs control loop cycle time).”
Page 8
*Figure 4. The grey zone extends over almost 150 ms according to the time axis unit. It is not easy to understand the link between this zone and the control loop of 500 microns (0.5 ms). This is about 1/300 with respect of the duration of the grey zone. An explanation will be helpful.
The text was changed to improve clarity. The ~140 us grey zone represents the time during which the reflectometry magnetic radius estimation (derived from the reflectometry outer separatrix estimation) replaced the actual magnetic radius measurement as the input of the radial position controller.
* Section 5. There is an issue with the use of the past. You are talking about events and things planned in the past for future even though they have been abandoned or amended.
The text was revised to consistently use past tense and to properly refer what would be the implications of the result if the diagnostic was not descoped.
Page 9.
*Figure 6. Is it useful?
We modified figure 5 to indicate some of the parts that required detailed design work and removed figure 6. To improve clarity, the caption of figure 5 was also modified to include “…indicating some of the parts that required a detailed design namely, the 120 bend, the 90
bends, and the antennas”.
Page 10.
*Figure 8. Is it useful? The shown information can be explained with some words in the text.
We agree, figure 8 has been removed from the manuscript.
Page 12.
*Figure 10. What do C_1, C_2, C_3 stand for?
C1, C2,.…, C9 stand for variations of the hyperbolic secant bends obtained with different parameters.
*MWS and HFSS. What are the main differences between these methods? How accurate are their results?
The main difference between the simulations conducted with MWS and HFSS is the type of solver used with each tool: the time-domain solver in MWS and the frequency-domain solver in HFSS. The fact that the simulations were conducted with two well-known and well-proved 3D EM simulators that use two completely different solvers provided a very good base for cross-benchmark of the results.
A reference to the type of solver used with each tool has been added to the text.
Page 14
*Figure 13. what do numbers stand for?
The numbers stand for the identification of the various prototype samples provided by the manufacturer.
The meaning of the numbers has been added to the text.
Page 15.
*Figure 14. Top panel, you may add male and female terms. The bottom panel is not clear for me.
Labels added. The bottom panel illustrates the setup used for the measurements. First, the reference isolation measurement is performed with both ports of the VNA directly terminated by a matched load. Secondly, the crosstalk measurement (A→D) is performed with the flanges inserted and B and C terminated using matched loads.
This clarification has been added to the text.
*Figure 15. What is the explanation of the discontinuity around 50 GHz?
The discontinuity is due to the different reference levels of the VNA in the 50-75 GHz band.
*L470. FTDT-->FDTD
Corrected.
Page 16.
* Figures 16 and 17. Are they necessary? Both of them?
Yes. Figure 16 describes the relevant aspects of the mock-up of the blanket modules used in the laboratory tests performed to assess the performance of gap 6 and we think it is necessary. Figure 17 is illustrative of how the complete setup was assembled in the anechoic chamber to measure the influence of the blanket modules in the antennas radiation pattern.
Page 17.
*L524-527. A bit complicated to understand. Higher frequencies ? which ones?
The (range of) frequencies corresponding to the “lower end of the frequency range” and to the “higher frequencies” are now specifically given in the text.
Page 18.
*L528-535. It is better to indicate on the figures these lobes.
The text was modified to make it clear that side-lobes correspond to radiation angles and that back-lobes correspond to radiation angles
.
Page 19.
IO-->ITER Organization (IO)
Done.
Page 20
*L623 [21][22][23]-->[21-23]
Done.
*L626. maximum precision of 5%? (minimum precision 5% instead, no?).
Thank you for spotting the incorrection. The correct is “maximum precision ERROR of 5%”. It was corrected in the paper
Page 21.
*L655. [23][27]-->[23,27]
Done.
*L665. CAD?
Corrected to “Computer-Aided Design (CAD)”.
Page 26
*Figure 26. There is a missing interpretation of the shown pictures.
The text regarding the picture was revised and further explanation added
Pages 27-28
*Ref [38] is not cited.
Thank you for spotting the incorrection. The reference was out of place. It is now reference [29] and cited in the text.
*Figure 28. Is this figure important to keep?
We consider that this figure shows that the prototype exists, it has a compact form factor and is already being tested in our facilities
*L912. Where is and who is developing this new prototype? When would it be ready (in few months?).
IPFN-IST was mentioned on the text but was not sufficiently clear. The sentence was rewritten to make it clear: “For that reason Instituto de Plasmas e Fusão Nuclear at Instituto Superior Técnico (IPFN-IST) in Lisbon, is developing and prototyping a coherent fast frequency sweeping RF back-end using commercial Monolithic Microwave Integrated Circuits (MMICs) leveraging the large-scale availability of high performance MMIC at affordable prices.”.
Regarding when it would be ready was added a sentence “The system is expected to be ready for deployment into experimental devices by early 2024.
Page 29.
*Ref [42] is cited before ref [41]. To permute.
Done
*L954. What the ITER PPR system was descoped? Some words would be very useful to understand. Now you are talking about DEMO.
We added a paragraph that explains the situation. The work of the consortium lead by IST was progressing according to schedule and we disagreed with the decision taken by the ITER Director. We would prefer to refrain to say more from what is written in the sentences below:
“ The ITER PPR system would allow to test the use of this diagnostic for control of a burning plasma, assess its performance under harsh conditions and to evaluate the design decisions taken and how those would affect the performance of the diagnostic. The R&D and design of the in-vessel components progressed according to schedule. Unfortunately, ITER terminated the PPR project due to concerns that captive components (waveguides and supports needed to install the system between the ports and the diagnostic hall) would not be procured and delivered in time. This would have resulted in unacceptable delays in installing the remaining captive components from other systems that could only be installed after the PPR system components were fully in place. As a result, ITER opted to rely on alternative technologies for measuring plasma position.”

Reviewer 2 Report
The author presents the microwave reflectometer for real-time measurements of plasma position and shape on fusion facilities. It is essential for fusion energy operation safety. I could see the design and lab measurements details on microwave reflectometer, but barely see the "control" function, which author mentioned in the title. So I recommend to replace that with ' measurements'.
(1) The author choose O-mode for reflectometer measurement, which has very delicate work to change transmitter/receiver antenna polarization orientation according to different magnetic pinch angle on the plasma edge. The different plasma current and toroidal magnetic field gives different pinch angle. The miss coupling will brings impurity on reflection signal. I believe the author has finished the calibration before installation, but missing make clear statement in this manuscript. Please add one paragraph about your polarization optimization.
(2) On page 6, line 232, the author provides the 1.5 cm and 2 cm offsets, which are very good numbers. Please provide the reflectometer radial resolution.
(3) Since the low frequency O-mode is chosen for reflectometer, we have to face the challenge that the shining spot in plasma is >= 3 cm, which might be a potential missing normal incident shining if plasma has vertical shift. I don't see any discuss about this in section 3.
(4) Please provide the reference about the control loop ~ 500 us, line 281.
(5) Please add sub-figure (a) and (b) on figure 4. Also please add colorbar on figure 4b.
(6) Please add reference on FTDT , line 470.
(7) Figure 18, I could see the horn gain is lower than 20 dB (frequency < 55 GHz). And the side lobe gain suppression is weak, for example, it is just 5 dB lower than main lobe on 22.25 GHz. That leads the receiver couples interference from oblique view (not normal incident). I could not agree this horn is qualified for reflectometer running in low frequency band (< 35 GHz).
(8) The author said noise level 2% , line 626. Please define the noise level and provide the calculation.
(9) On figure 22, there will be multiple groups reflectometers occupied. Will all reflectometers run simultaneously as FMCW mode? If so, the cross-interference will be great challenge, since low gain horn, non-normal incident angle, wall reflection, etc. The in-vessel global microwave pollution has been clearly observed on DIII-D and other facilities. The author also provide the non-local beam coupling on figure 26. Please provide the potential solution or discussion to avoid this serious problem.
(10) The compact reflectometer is an innovative development, which will be the most popular solution for fusion plasma measurements. But we have to face two major challenges, (i) high performance with excellent shielding against X-ray, Gamma, neutron; (ii) affordable price. I could see the prototype development with commercial circuits components on customized PCB base. It is one possible solution, but outdated. The most promise solution for high level integration microwave diagnostics is system-on-chip, which develop all microwave circuits on 10 mm2 semi-conductor chip (CMOS, InP, GaAs, and GaN). This technology has been successfully developed on DIII-D since 2019. This work is also presented on the conferences, including APS, ECPD, HTPD. I agree with the author, the compact microwave diagnostics is essential. So I recommend the development should go with system-on-chip.
Author Response
We would like to thank the reviewers for the comments that helped to improve the readability of the paper. Below you can find a detailed reply to the reviewers regarding the changes introduced.
We opted to follow the suggestion of reviewer 1 and remove some figures. In some cases the relevant information was added to the other figures or described in the text. Minor rearrangements of the figures were done to make better use of the space available.
The author presents the microwave reflectometer for real-time measurements of plasma position and shape on fusion facilities. It is essential for fusion energy operation safety. I could see the design and lab measurements details on microwave reflectometer, but barely see the "control" function, which author mentioned in the title. So I recommend to replace that with 'measurements'.
Thank you for the comment. Although it is true that we are performing real-time measurements the final goal is to control the plasma position and shape and the design must be driven by that goal. Furthermore, the measurements were performed, and actually used, to control the plasma in ASDEX-Upgrade and COMPASS. All the remaining developments are performed with the aim to be able to control the plasma in mind. We consider that the title mentioning Control keeps the focus on the goal of the development.
(1) The author chooses O-mode for reflectometer measurement, which has very delicate work to change transmitter/receiver antenna polarization orientation according to different magnetic pinch angle on the plasma edge. The different plasma current and toroidal magnetic field gives different pinch angle. The miss coupling will bring impurity on reflection signal. I believe the author has finished the calibration before installation, but missing make clear statement in this manuscript. Please add one paragraph about your polarization optimization.
Thank you for the comment. The FMCW microwave reflectometry system on ASDEX Upgrade uses fundamental rectangular waveguides connected to hog-horns with elliptical mirrors. Their orientation sets the electrical field of the probing beam in the toroidal direction. The ASDEX Upgrade magnetic pitch angle is in the order of 10º [0] which implies that 97% of the probing signal is sent in O-mode. The 3% send in X-mode is attenuated in the waveguide. Therefore, no calibration procedure is required to set the correct polarization. If circular waveguides were used the correct polarization would require calibration using a mirror with a grid. This discussion was added to the manuscript
[1] D. Prisiazhniuk et al., Plasma Physics and Controlled Fusion,59 (2) (2017) 025013, https://dx.doi.org/10.1088/1361-6587/59/2/025013
(2) On page 6, line 232, the author provides the 1.5 cm and 2 cm offsets, which are very good numbers. Please provide the reflectometer radial resolution.
O-mode full-wave 1D simulations were performed to assess measurement sensitivity to plasma turbulence and initialization of the non-probed plasma density range (0£ne£0.36 x 1019 m-3) [1]. These simulations demonstrated that the spatial resolution remains always below 5 mm. This information and the reference was added to the text
[1] Santos, J., Hacquin, S., & Manso, M. (2004). Frequency modulation continuous wave reflectometry measurements of plasma position in ASDEX Upgrade ELMy H-mode regimes. Review of Scientific Instruments, 75(10), 3855–3858. https://doi.org/DOI:10.1063/1.1788832
(3) Since the low frequency O-mode is chosen for reflectometer, we have to face the challenge that the shining spot in plasma is >= 3 cm, which might be a potential missing normal incident shining if plasma has vertical shift. I don't see any discuss about this in section 3.
The FMCW microwave reflectometry system on ASDEX Upgrade is installed at the equatorial plane and each channel uses a single antenna to emit and receive (mono-static), where the reception is always optimised with moderated directivity. The system proved to cope quite well with rather large vertical displacements in the order of 20 cm [Erro! A origem da referência não foi encontrada.]. This information was added to the text
[2] A. Silva, PhD thesis (2007), pg 86, https://www.researchgate.net/publication/259272611_The_ASDEX_Upgrade_broadband_microwave_reflectometry_system
(4) Please provide the reference about the control loop ~ 500 us, line 281.
Introduced reference:
F. Janky, J. Havlicek, A. J. N. Batista et al., “Upgrade of the COMPASS tokamak real-time
control system,” Fusion Engineering and Design, vol. 89, no. 3, pp. 186–194, 2014, DOI:
10.1016/j.fusengdes.2013.12.042
(5) Please add sub-figure (a) and (b) on figure 4. Also please add color bar on figure 4b.
The figure was updated and the color bar added
(6) Please add reference on FTDT , line 470.
We added 3 references we consider relevant in this context:
- The technique has been delineated in: Kane Yee. (1966). Numerical solution of initial boundary value problems involving maxwell's equations in isotropic media. Antennas and Propagation, IEEE Transactions on, 14(3), 302–307. http://doi.org/10.1109/TAP.1966.1138693.
- The diffusion and use of the term FDTD was done by: Taflove, A. & Hagness, S. C. (2005). Computational Electrodynamics: The Finite-Difference Time-Domain Method, Third Edition (3rd ed.). Artech House.
- The application of FDTD for reflectometry was established in: da Silva, F., Pinto, M. C., Després, B., & Heuraux, S. (2015). Stable explicit coupling of the Yee scheme with a linear current model in fluctuating magnetized plasmas, 295, 24–45. http://doi.org/10.1016/j.jcp.2015.03.069.
(7) Figure 18, I could see the horn gain is lower than 20 dB (frequency < 55 GHz). And the side lobe gain suppression is weak, for example, it is just 5 dB lower than main lobe on 22.25 GHz. That leads the receiver couples interference from oblique view (not normal incident). I could not agree this horn is qualified for reflectometer running in low frequency band (< 35 GHz).
It is well-known that small-aperture pyramidal horns are not well suited for low-frequency measurements due to the broad radiation pattern they exhibit at these frequencies. Because the setup of gap 6 was very complex, with the antennas placed between blanket modules, the ability of the system to perform reflectometry measurements within the required error margin (1 cm) was assessed by laboratory tests with a target mirror positioned at various distances from the antennas. As stated in the text, the system was found to perform as intended except for frequencies below 26.5 GHz (K band), when the mirror is located closer to the antennas. In this band, the frequency below which the errors are larger than
1 cm decrease as the distance to the mirror increases, being as low as 18 GHz for a distance of 35 cm. By lapse, reference [16] was not cited in the text. This has now been corrected.
(8) The author said noise level 2% , line 626. Please define the noise level and provide the calculation.
The 2% noise level is a requirement from the “DEMO Diagnostics and Control (DC) System Requirements Document (SRD)”(reference cited in the text [1] ) It is a target that we need to try to achieve by design for the Electron Density Profile (ne) in the pedestal region using microwave reflectometry.
[1] Jesenko, A. et al., Diagnostics and Control (DC) System Requirements Document (SRD), EUROfusion report IDM: EFDA_D_2MNK4R_v3.0 (2020)).
(9) On figure 22, there will be multiple groups reflectometers occupied. Will all reflectometers run simultaneously as FMCW mode? If so, the cross-interference will be great challenge, since low gain horn, non-normal incident angle, wall reflection, etc. The in-vessel global microwave pollution has been clearly observed on DIII-D and other facilities. The author also provides the non-local beam coupling on figure 26. Please provide the potential solution or discussion to avoid this serious problem.
Thank you for highlighting this challenge. We added the following discussion to the paper “With all reflectometers running simultaneously in FMCW mode the cross-interference will be challenging, due low gain horn, non-normal incident angle, wall reflection, etc. This in-vessel global microwave pollution has been clearly observed on DIII-D and other facilities. Two approaches can be followed: a) synchronized sweeping of all reflectometers with a time shift to assure that there are no reflectometers at the same frequency at a given time; b) Use Direct Digital Synthesis (DDS) but instead of using a linear sweep, use an encoded sweep with a different code for each reflectometer and only the received signals that show high correlation with the transmitted one will be selected. Lot of these technics are being developed for the automotive industry to overcome a similar problem of interference between the radars of different vehicles that share the same road.”
(10) The compact reflectometer is an innovative development, which will be the most popular solution for fusion plasma measurements. But we have to face two major challenges, (i) high performance with excellent shielding against X-ray, Gamma, neutron; (ii) affordable price. I could see the prototype development with commercial circuits components on customized PCB base. It is one possible solution, but outdated. The most promise solution for high level integration microwave diagnostics is system-on-chip, which develop all microwave circuits on 10 mm2 semi-conductor chip (CMOS, InP, GaAs, and GaN). This technology has been successfully developed on DIII-D since 2019. This work is also presented on the conferences, including APS, ECPD, HTPD. I agree with the author, the compact microwave diagnostics is essential. So I recommend the development should go with system-on-chip.
Thank you for the very relevant comment. We are aware that several reflectometers on a chip have been used on DIII-D but with application to imaging reflectometry where fixed frequency or very limited frequency sweep is used. The presented prototype could have been produced with a larger level of integration but in this way, it was possible to better optimize the different stages of the reflectometer. Radar on a chip exists for massive applications in the automotive industry but are bandwidth limited. I should note that the US DA doesn’t propose the “on a chip” solution for the ITER main plasma reflectometers. But I agree with you that the future will lead to a large integration system on a chip or a couple of chips. A discussion on this matter on included in the text (making use of part of the information provided by the reviewer)

Reviewer 3 Report
The paper presents a comprehensive review of microwave reflectometry diagnosis of fusion plasma, as well as perspectives and challenges for further improving this technique to use it in the future fusion devices like ITER and DEMO. The article is well written and organized and the presented information is useful for researchers involved in fusion science and not only.
There are some minor problems, related only to the article editing:
1. At the "Author Contributions" section (page 29, rows 975-976), please delete the indication: "For research articles with several authors, a short paragraph specifying their individual contributions must be provided. The following statements should be used".
2. If you have nothing to declare at the "Acknowledgements" section, please delete it.
3. In the Reference section there are several problems:
- first of all, according to my knowledge, the reference format used by you is wrong, since the journal asks, for example, the name of the authors written as "Santos, J." and not "J. Santos";
- at reference [2], please delete "R. Sabot 2006" and keep only "Sabot, R. et al. ...";
- references [5] and [11] are identical. When you will remove the reference [11], please pay attention in the whole article to the references numbers;
- for Journal of Instrumentation the standard abbreviation is J. Instrum. and not JINST as you used (references [17], [20], [26], [37] and [39]);
- at the reference [25] please use the abbreviation IEEE Trans. Plasma Sci.
Author Response
We would like to thank the reviewers for the comments that helped to improve the readability of the paper. Below you can find a detailed reply to the reviewers regarding the changes introduced.
We opted to follow the suggestion of reviewer 1 and remove some figures. In some cases the relevant information was added to the other figures or described in the text. Minor rearrangements of the figures were done to make better use of the space available.
The paper presents a comprehensive review of microwave reflectometry diagnosis of fusion plasma, as well as perspectives and challenges for further improving this technique to use it in the future fusion devices like ITER and DEMO. The article is well written and organized and the presented information is useful for researchers involved in fusion science and not only.
There are some minor problems, related only to the article editing:
1. At the "Author Contributions" section (page 29, rows 975-976), please delete the indication: "For research articles with several authors, a short paragraph specifying their individual contributions must be provided. The following statements should be used".
Done.
2. If you have nothing to declare at the "Acknowledgements" section, please delete it.
Done
3. In the Reference section there are several problems:
- first of all, according to my knowledge, the reference format used by you is wrong, since the journal asks, for example, the name of the authors written as "Santos, J." and not "J. Santos".
Done.
- at reference [2], please delete "R. Sabot 2006" and keep only "Sabot, R. et al. ...".
Done.
- references [5] and [11] are identical. When you will remove the reference [11], please pay attention in the whole article to the references numbers.
Done.
- for Journal of Instrumentation the standard abbreviation is J. Instrum. and not JINST as you used (references [17], [20], [26], [37] and [39]).
Done.
- at the reference [25] please use the abbreviation IEEE Trans. Plasma Sci.
Done.

Round 2
Reviewer 1 Report
The authors have amended the manuscript according to the major comments of my report. I think this new version is satisfying even if it can still be improved. However it can be accepted for publication in its present form.
Reviewer 2 Report
Thanks to author carefully improvements. It looks much better than previous. Congratulations to your excellent work. Last comment, I heard the chip development report on high temperature plasma diagnostic conference, that the imaging reflectometer chip could run on FMCW mode and fixed frequency mode both.